# Causal normalizing flows: from theory to practice

**Adrián Javaloy** [1,*]   **Pablo Sánchez-Martín** [1,2]   **Isabel Valera** [1,3]

[1]Department of Computer Science of Saarland University, Saarbrücken, Germany
[2]Max Planck Institute for Intelligent Systems, Tübingen, Germany
[3]Max Planck Institute for Software Systems, Saarbrücken, Germany

## Abstract

In this work, we deepen on the use of normalizing flows for causal inference. Specifically, we first leverage recent results on non-linear ICA to show that causal models are identifiable from observational data given a causal ordering, and thus can be recovered using autoregressive normalizing flows (NFs). Second, we analyse different design and learning choices for *causal normalizing flows* to capture the underlying causal data-generating process. Third, we describe how to implement the *do-operator* in causal NFs, and thus, how to answer interventional and counterfactual questions. Finally, in our experiments, we validate our design and training choices through a comprehensive ablation study; compare causal NFs to other approaches for approximating causal models; and empirically demonstrate that causal NFs can be used to address real-world problems—where mixed discrete-continuous data and partial knowledge on the causal graph is the norm. The code for this work can be found at https://github.com/psanch21/causal-flows.

## 1   Introduction

Deep learning is increasingly used for causal reasoning, that is, for finding the underlying causal relationships among the observed variables (*causal discovery*), and answering *what-if* questions (*causal inference*) from available data [28]. Our focus in this paper is to solve causal inference problems using only observational data and (potentially partial) knowledge on the causal graph of the underlying structural causal model (SCM). This is exemplified in Fig. 1, where our proposed framework is able to estimate the (unobserved) causal effect of externally intervening on the sensitive attribute (red and yellow distributions), *using solely observed data (blue distribution) and partial information about the causal relationship between features*.

In this context, previous works have mostly relied on different deep neural networks (DNNs)—e.g., nor-

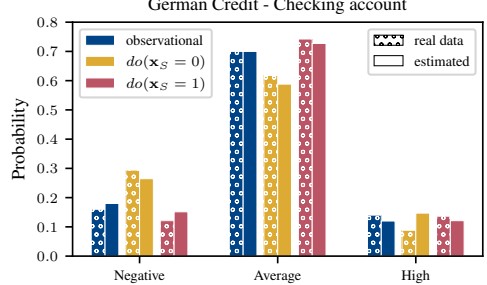

Figure 1: Observational and interventional distributions of the categorical variable *checking account* of the German Credit dataset [8], and their estimated values according to a causal normalizing flow. $\mathbf{x}_S$ is a binary variable representing the users' sex.

malizing flows (NFs) [23, 26, 27], generative adversarial networks (GANs) [20, 39], variational autoencoders (VAEs) [15, 40], Gaussian processes (GPs) [15], or denoising diffusion probabilistic models (DDPMs) [2]—to iteratively estimate the conditional distribution of each observed variable given its causal parents, thus using an independent DNN per observed variable. Hence, to predict the effect of an intervention in the causal data-generating process, these approaches fix the value of the intervened variables when computing the new value for their children. However, they may also suffer

---

*Correspondence to ajavaloy@cs.uni-saarland.de.

37th Conference on Neural Information Processing Systems (NeurIPS 2023).

from error propagation—which worsens with long causal paths—and a high number of parameters, which is addressed in practice with ad-hoc parameter amortization techniques [26, 27]. Moreover, several approaches also rely on implicit distributions [2, 20, 27, 32], and thus do not allow evaluating the learnt distribution.

In contrast, and similar to [16, 32, 34, 41], we here aim at learning the full causal-generating process using a single DNN and, in particular, using a *causal normalizing flow*. To this end, we first theoretically demonstrate that causal NFs are a natural choice to approximate a broad class of causal data-generating processes (§3). Then, we design causal NFs that inherently satisfy the necessary conditions to capture the underlying causal dependencies (§4), and introduce an implementation of the do-operator that allows us to efficiently solve causal inference tasks (§5). Importantly, our causal NF framework allows us to deal with mixed continuous-discrete data and partial knowledge on the causal graph, which is key for real-world applications. Finally, we empirically validate our findings and show that causal NFs outperform competing methods also using a single DNN to approximate the causal data-generating process (§6).

**Related work**  To the best of our knowledge, the closest works to ours are [16, 34, 41], as they all capture the whole causal data-generating process using a single DNN. Our approach generalizes the result from Khemakhem et al. [16], which also relies on autoregressive normalizing flows (ANF), but only considers affine ANFs and data with additive noise. In contrast, our work provides a tighter connection between ANFs and SCMs (affine or not), more general identifiability results, and sound ways to both embed causal knowledge in the ANF, and to apply the do-operator. Another relevant line of works connect SCMs with GNNs [32, 41], and despite making little assumptions on the underlying SCM, they lack identifiability guarantees, and interventions on the GNN are performed by severing the graph, which we show in App. C may not work in general. Nevertheless, it is worth noting that the way we use $\boldsymbol{A}$ in the network design (§4) is inspired by these works.

## 2   Preliminaries and background

### 2.1   Structural causal models, interventions, and counterfactuals

A structural causal model (SCM) [28] is a tuple $\mathcal{M} = (\tilde{\mathbf{f}}, P_{\mathbf{u}})$ describing a data-generating process that transforms a set of $d$ exogenous random variables, $\mathbf{u} \sim P_{\mathbf{u}}$, into a set of $d$ (observed) endogenous random variables, $\mathbf{x}$, according to $\tilde{\mathbf{f}}$. Specifically, the endogenous variables are computed as follows:

$$\mathbf{u} := (\mathbf{u}_1, \mathbf{u}_2, \ldots, \mathbf{u}_d) \sim P_{\mathbf{u}}, \qquad \mathbf{x}_i = \tilde{f}_i(\mathbf{x}_{\mathrm{pa}_i}, \mathbf{u}_i), \qquad \text{for } i = 1, 2, \ldots, d. \qquad (1)$$

In other words, each $i$-th component of $\tilde{\mathbf{f}}$ maps the $i$-th exogenous variable $\mathbf{u}_i$ to the $i$-th endogenous variable $\mathbf{x}_i$, given the subset of the endogenous variables that directly cause $\mathbf{x}_i$, $\mathbf{x}_{\mathrm{pa}_i}$ (causal parents).

An SCM also induces a causal graph, a powerful tool to reason about the causal dependencies of the system. Namely, the causal graph of an SCM $\mathcal{M} = (\tilde{\mathbf{f}}, P_{\mathbf{u}})$ is the directed graph that describes the functional dependencies of the causal mechanism. We can define the *adjacency matrix of the causal graph* as $\boldsymbol{A} := \nabla_{\mathbf{x}} \tilde{\mathbf{f}}(\mathbf{x}, \mathbf{u}) \neq \mathbf{0}$, where $\mathbf{0}$ is the constant zero function, and the comparisons are made elementwise. Furthermore, the direct causes of the $i$-th variable ($\mathrm{pa}_i$ in Eq. 1) are the *parent* nodes of the $i$-th node in $\boldsymbol{A}$, and the *ancestors* of this node (which we denoted by $\mathrm{an}_i$) are its (in)direct causes. See Fig. 2 for an example of a causal chain.

$$\text{x}_1 \longrightarrow \text{x}_2 \longrightarrow \text{x}_3$$

$$\pi = \begin{pmatrix} 1 & 2 & 3 \end{pmatrix}$$

$$\boldsymbol{A} = \begin{pmatrix} 0 & 0 & 0 \\ 1 & 0 & 0 \\ 0 & 1 & 0 \end{pmatrix}$$

Figure 2: Causal graph, and its causal ordering $\pi$ and adjacency matrix $\boldsymbol{A}$.

In the case that $\boldsymbol{A}$ is acyclic, we can pick a causal ordering describing which variables do *not* cause others, and which ones *may* cause them. Namely, a permutation $\pi$ is said to be a *causal ordering* of an SCM $\mathcal{M}$ if, for every $\mathbf{x}_i$ that directly causes $\mathbf{x}_j$, we have $\pi(i) < \pi(j)$. Note that this definition equals that of a topological ordering and, without loss of generality, we will assume throughout this work that the variables are sorted according to a causal ordering.

Importantly, besides describing the (observational) data-generating process, SCMs enable *causal inference* by allowing us to answer *what-if* questions regarding: i) how the distribution over the observed variables would be if we force a fixed value on one of them (*interventional queries*); and ii) what would have happened to a specific observation, if one of its dimensions would have taken a different value (*counterfactual queries*).

**Structural equivalence**  To reason about causal dependencies, we introduce the notion of structural equivalence. We say that two matrices $S$ and $R$ are *structurally equivalent*, denoted $S \equiv R$, if both matrices have zeroes exactly in the same positions. Similarly, we say that $S$ is *structurally sparser* than $R$, denoted as $S \preceq R$, if whenever an element of $R$ is zero, the same element of $S$ is zero.

## 2.2  Autoregressive normalizing flows

Normalizing flows (NFs) [25] are a model family that express the probability density of a set of observations using the change-of-variables rule. Given an observed random vector $\mathbf{x}$ of size $d$, a normalizing flow is a neural network with parameters $\boldsymbol{\theta}$ that takes $\mathbf{x}$ as input, and outputs

$$T_{\boldsymbol{\theta}}(\mathbf{x}) =: \mathbf{u} \sim P_{\mathbf{u}} \quad \text{with log-density} \quad \log p(\mathbf{x}) = \log p(T_{\boldsymbol{\theta}}(\mathbf{x})) + \log|\det(\nabla_{\mathbf{x}} T_{\boldsymbol{\theta}}(\mathbf{x}))|, \quad (2)$$

where $P_{\mathbf{u}}$ is a base distribution that is easy to evaluate and sample from. Since Eq. 2 provides the log-likelihood expression, it naturally leads to the use of maximum likelihood estimation (MLE) [1] for learning the network parameters $\boldsymbol{\theta}$. While many approaches have been proposed in the literature [25], here we focus on autoregressive normalizing flows (ANFs) [18, 24]. Specifically, in ANFs the $i$-th output of each layer $l$ of the network, denoted by $z_i^l$, is computed as

$$z_i^l := \tau_i^l(z_i^{l-1}; \mathbf{h}_i^l), \quad \text{where} \quad \mathbf{h}_i^l := c_i^l(\mathbf{z}_{1:i-1}^{l-1}), \quad (3)$$

and where $\tau_i$ and $c_i$ are termed the transformer and the conditioner, respectively. The transformer is a strictly monotonic function of $z_i^{l-1}$, while the conditioner can be arbitrarily complex, yet it only takes the variables preceding $z_i$ as input. As a result, ANFs have triangular Jacobian matrices, $\nabla_{\mathbf{x}} T_{\boldsymbol{\theta}}(\mathbf{x})$.

# 3  Causal normalizing flows

**Problem statement**  Assume that we have a sequence of i.i.d. observations $\boldsymbol{X} = \{\boldsymbol{x}_1, \boldsymbol{x}_2, \ldots, \boldsymbol{x}_N\}$ generated according to an unknown SCM $\mathcal{M}$, from which we have partial knowledge of its causal structure. Specifically, we know at least its causal ordering $\pi$, and at most the whole causal graph $\boldsymbol{A}$. Our objective in this work is to design and learn an ANF $T_{\boldsymbol{\theta}}$, with parameters $\boldsymbol{\theta}$, that captures $\mathcal{M}$ by maximizing the observational likelihood (MLE), i.e.,

$$\underset{\boldsymbol{\theta}}{\text{maximize}} \frac{1}{N} \sum_{n=1}^{N} \Big[ \log p\left(T_{\boldsymbol{\theta}}\left(\boldsymbol{x}_n\right)\right) + \log|\det(\nabla_{\mathbf{x}} T_{\boldsymbol{\theta}}\left(\boldsymbol{x}_n\right))| \Big], \quad (4)$$

and that can successfully answer interventional and counterfactual queries during deployment, thus enabling causal inference. We refer to such a model as a *causal normalizing flow*.

**Assumptions**  We restrict the class of SCMs considered by making the following fairly common assumptions: i) *diffeomorphic data-generating process*, i.e., $\tilde{\mathbf{f}}$ is invertible, and both $\tilde{\mathbf{f}}$ and its inverse are differentiable; ii) *no feedback loops*, i.e., the induced causal graph is acyclic; and iii) *causal sufficiency*, i.e., the exogenous variables are mutually independent, $p(\mathbf{u}) = \prod_i p(\mathbf{u}_i)$.

**SCMs as TMI maps**  To achieve our objective, and bridge the gap between SCMs and normalizing flows, we resort to triangular monotonic increasing (TMI) maps, which are autoregressive functions whose $i$-th component is strictly monotonic increasing with respect to its $i$-th input. TMI maps hold a number of useful properties, such as being closed under composition and inversions. Conveniently, a layer of an ANF (Eq. 3) is a parametric TMI map that can approximate any other TMI map arbitrarily well, which makes ANFs also TMI maps approximators;[2] a fact that has been exploited in the past to prove that ANFs are universal density approximators [25].

We now show that any SCM can be rewritten as a tuple $(\mathbf{f}, P_{\mathbf{u}}) \in \mathcal{F} \times \mathcal{P}_{\mathbf{u}}$, where $\mathcal{F}$ is the set of all TMI maps, and $P_{\mathbf{u}}$ is the set of all fully-factorized distributions, $p(\mathbf{u}) = \prod_i p(\mathbf{u}_i)$. First, given an acyclic SCM $\mathcal{M} = (\tilde{\mathbf{f}}, P_{\mathbf{u}})$ with $\tilde{\mathbf{f}} : \mathbb{X} \times \mathbb{U} \to \mathbb{X}$ as in Eq. 1, we can always unroll $\tilde{\mathbf{f}}$ by recursively replacing each $x_i$ in the causal equation by its function $\tilde{f}_i$ (see Fig. 4b for an example), obtaining an equivalent non-recursive function $\hat{\mathbf{f}} : \mathbb{U} \to \mathbb{X}$. This function $\hat{\mathbf{f}}$ writes each $x_i$ as a function of its exogenous ancestors $\mathbf{u}_{\text{an}_i}$ and, since $\mathcal{M}$ is acyclic, $\hat{\mathbf{f}}$ is a triangular map. For simplicity, assume that $P_{\mathbf{u}}$ is a standard uniform distribution. Then, following the causal ordering, we can apply a Darmois

---

[2]While it is a common to shuffle the inputs for each layer, we keep the same order across the network.

construction [7, 13] and replace each function $\hat{f}_i$ by the conditional quantile function of the variable $\mathrm{x}_i$ given $\mathbf{x}_{\mathrm{pa}_i}$ (which depends on $\mathbf{u}_{\mathrm{an}_i}$) eventually arriving to a TMI map $\mathbf{f}$. This procedure follows the proof for non-identifiability in ICA [13], but restricted to one ordering. The case for a general $P_{\mathbf{u}}$ follows a similar construction, but using a Knöthe-Rosenblatt (KR) transport [19, 31] instead.

**Isolating the exogenous variables**   Now that we have SCMs and causal NFs under the same family class—i.e., the family $\mathcal{F} \times \mathcal{P}_{\mathbf{u}}$ of TMI maps with fully-factorized distributions—we leverage existing results on identifiability to show that we can find a causal NF $T_{\boldsymbol{\theta}}$ such that the $i$-th component of $T_{\boldsymbol{\theta}}(\mathbf{x})$ is a function of the true exogenous variable $\mathrm{u}_i$ that generated the observed data. More precisely, note that, since we can rewrite $\mathcal{M}$ as an element of the family $\mathcal{F} \times \mathcal{P}_{\mathbf{u}}$, identifying the true exogenous variables of an SCM $\mathcal{M}$ is equivalent to solving a non-linear ICA problem with TMI generators, for which Xi and Bloem-Reddy [38] proved the following (re-stated to match our setting):

**Theorem 1** (Identifiability). *If two elements of the family $\mathcal{F} \times \mathcal{P}_{\mathbf{u}}$ (as defined above) produce the same observational distribution, then the two data-generating processes differ by an invertible, component-wise transformation of the variables $\mathbf{u}$.*

Thm. 1 implies that, if we can find a causal NF $(T_{\boldsymbol{\theta}}, P_{\boldsymbol{\theta}}) \in \mathcal{F} \times \mathcal{P}_{\mathbf{u}}$ that matches the observational distribution generated by $\mathcal{M} = (\mathbf{f}, P_{\mathcal{M}}) \in \mathcal{F} \times \mathcal{P}_{\mathbf{u}}$, then we know that the exogenous variables of the flow differ from the real ones by a function of each component independently, i.e., $T_{\boldsymbol{\theta}}(\mathbf{f}(\mathbf{u})) \sim P_{\boldsymbol{\theta}}$ with $\mathbf{u} \sim P_{\mathcal{M}}$ and $T_{\boldsymbol{\theta}}(\mathbf{f}(\mathbf{u})) = \boldsymbol{h}(\mathbf{u}) = (h_1(\mathbf{u}_1), h_2(\mathbf{u}_2), \ldots, h_d(\mathbf{u}_d))$, where each $h_i$ is an invertible function. Fig. 3 graphically illustrates Thm. 1. Furthermore, Thm. 1 also implies that the functional dependencies of the causal NF must agree with that of the SCM, i.e., that $T_{\boldsymbol{\theta}}$ needs to be *causally consistent* with $\mathcal{M}$. We formally present this result in the following corollary (proof can be found in App. A), where $\boldsymbol{I}$ denotes the identity matrix:

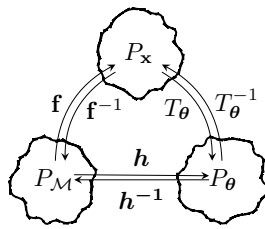

Figure 3: Thm. 1 as a commutative diagram

**Corollary 2** (Causal consistency). *If a causal NF $T_{\boldsymbol{\theta}}$ isolates the exogenous variables of an SCM $\mathcal{M}$, then $\nabla_{\mathbf{x}} T_{\boldsymbol{\theta}}(\mathbf{x}) \equiv \boldsymbol{I} - \boldsymbol{A}$ and $\nabla_{\mathbf{u}} T_{\boldsymbol{\theta}}^{-1}(\mathbf{u}) \equiv \boldsymbol{I} + \sum_{n=1}^{\mathrm{diam}(\boldsymbol{A})} \boldsymbol{A}^n$, where $\boldsymbol{A}$ is the causal adjacency matrix of $\mathcal{M}$. In other words, $T_{\boldsymbol{\theta}}$ is causally consistent with the true data-generating process, $\mathcal{M}$.*

A sketch of the proof goes as follows: since Fig. 3 is a commutative diagram, we can write the result of $T_{\boldsymbol{\theta}}$ and $T_{\boldsymbol{\theta}}^{-1}$ in terms of the true $\mathbf{f}, \boldsymbol{h}$, and their inverses. Then, we can use the chain rule to compute their Jacobian matrices, and since $\boldsymbol{h}$ has a diagonal Jacobian matrix, it preserves the structure of the Jacobian matrices of $\mathbf{f}$ and its inverse. To sum up, we have shown that *causal NFs are a natural choice to estimate an unknown SCM* by showing that: i) both SCMs and causal NFs fall within the same family $\mathcal{F} \times \mathcal{P}_{\mathbf{u}}$; ii) any two elements of this family with identical observational distributions are causally consistent; and iii) they differ by an invertible component-wise transformation.

### 3.1   Causal NFs for real-world problems

To bring theory closer to practice, we need to extend causal NFs to handle mixed discrete-continuous data and partial knowledge on the causal graph, which are common properties of real-world problems. Due to space limitations, we provide here a brief explanation, and formalize these ideas in App. A.2.

**Discrete data**   To extend our results to also account for discrete data, we take advantage of the general model considered by Xi and Bloem-Reddy [38] that includes observational noise (independent of the exogenous variables), and consider a continuous version of the observed discrete variables by adding to them independent noise $\varepsilon \in [0, 1]$ (e.g., from a standard uniform), such that the real distribution is still recoverable. Intuitively, our approach assumes that discrete variables correspond to the integer part of (noisy) continuous variables generated according to an SCM fulfilling our assumptions, such that both our theoretical and practical insights still apply.

**Partial knowledge**   While we rarely know the entire causal graph $\boldsymbol{A}$, we often have a good grasp on causal relationships between a subset of observed variables—e.g., sex and age are not causally related—while missing the rest. When only partial knowledge on the graph is available—i.e., we only know the causal relationship between a subset of the observed variables, we can instead work with a modified acyclic graph $\tilde{\boldsymbol{A}}$ obtained by finding the strongly connected components as in [35], where subsets of variables with unknown causal relationships are treated as a block (see §7 for an example).

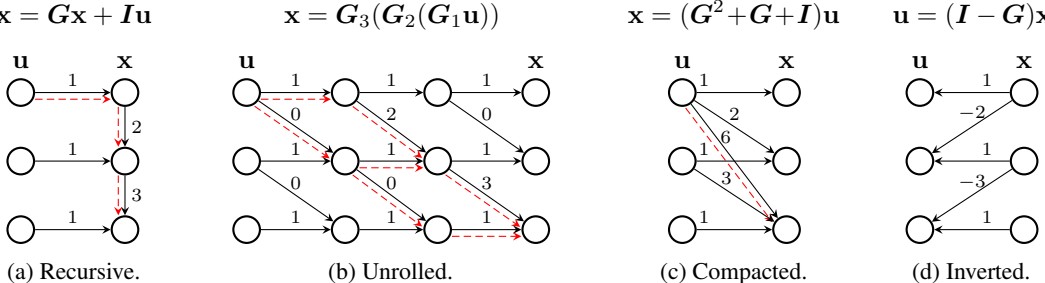

$$\mathbf{x} = \boldsymbol{G}\mathbf{x} + \boldsymbol{I}\mathbf{u} \qquad \mathbf{x} = \boldsymbol{G}_3(\boldsymbol{G}_2(\boldsymbol{G}_1\mathbf{u})) \qquad \mathbf{x} = (\boldsymbol{G}^2 + \boldsymbol{G} + \boldsymbol{I})\mathbf{u} \qquad \mathbf{u} = (\boldsymbol{I} - \boldsymbol{G})\mathbf{x}$$

(a) Recursive.        (b) Unrolled.        (c) Compacted.        (d) Inverted.

Figure 4: Example of the linear SCM $\{x_1 := u_1 ; x_2 := 2x_1 + u_2 ; x_3 := 3x_2 + u_3\}$ written (a) in its usual recursive formulation; (b) without recursions, with each step made explicit; (c) without recursions, as a single function; and (d) writing $\mathbf{u}$ as a function of $\mathbf{x}$. The red dashed arrows show the influence of $u_1$ on $x_3$ for all equations from $\mathbf{u}$ to $\mathbf{x}$, with the compacted version exhibiting shortcuts (see §4). Note that in the linear case we have $\boldsymbol{A} := \boldsymbol{G} \neq \mathbf{0}$, and that $\boldsymbol{G}_1, \boldsymbol{G}_2, \boldsymbol{G}_3 \preceq \boldsymbol{G} + \boldsymbol{I}$ are any three matrices such that their product equals $\boldsymbol{G}^2 + \boldsymbol{G} + \boldsymbol{I}$.

This allows us to reuse our theoretical results for known parts of the graph, thus generalizing the *block identifiability* results from von Kügelgen et al. [36].

## 4 Effective design of causal normalizing flows

We showed in §3 that causal NFs are a natural choice to learn the underlying SCM generating the data. Importantly, Thm. 1 assumes that we can find a causal NF whose observational distribution perfectly matches the true data distribution (according to the underlying SCM). In practice, however, reaching the optimal parameters may be tricky as: i) we only have access to a finite amount of training data; and ii) the optimization process for causal NFs (like for any neural network) may converge to a local optima. In this section, we analyse different design choices for causal NFs to guide the optimization towards solutions that do not only provide an accurate fit of the observational distribution, but allow us to also accurately answer to interventional and counterfactual queries.

Let us start with an illustrative example. Suppose that we are given the linear SCM in Fig. 4a, and we want to write the SCM equations as a TMI map to approximate them with a causal NF. As discussed in §3, we can unroll the causal equations (Fig. 4b)—resulting in a composition of functions structurally as sparse as $\boldsymbol{I} + \boldsymbol{A}$. These functions can be compacted into a single transformation (Fig. 4c), such that each $x_i$ depends on its ancestors, $\mathbf{u}_{\mathrm{an}_i}$. However, note that in this step *shortcuts* appear, making direct and indirect causal paths in this representation indistinguishable—in our example, the indirect causal path from $u_1$ to $x_3$ present in Fig. 4a and Fig. 4b does not go anymore through the path that generates $x_2$, but instead via a *shortcut* that directly connects $u_1$ to $x_3$. Alternatively, we can invert the equations to write $\mathbf{u}$ as a function of $\mathbf{x}$ (Fig. 4d), which is structurally equivalent to $\boldsymbol{I} - \boldsymbol{A}$.

We remark that the above steps can be applied to any considered acyclic SCM (refer to App. B for a more detailed discussion). In particular, we can unroll the equations in a finite number of steps, and we can similarly reason about the causal dependencies through the Jacobian matrices of the generators, $\nabla_{\mathbf{x}} T_{\boldsymbol{\theta}}(\mathbf{x})$ and $\nabla_{\mathbf{u}} T_{\boldsymbol{\theta}}^{-1}(\mathbf{u})$. Moreover, note that the diffeomorphic assumption implies that we can invert the causal equations. Next, inspired by the different representations of an SCM (exemplified in Fig. 4), we consider the following design choices for causal NFs:

**Generative model** The first architecture imitates the unrolled equations (Fig. 4b), i.e., the causal NF is defined as a function from $\mathbf{u}$ to $\mathbf{x}$. Importantly, when full knowledge of the causal graph $\boldsymbol{A}$ is assumed, we also replicate the structural sparsity per layer by adequately masking the flow with $\boldsymbol{I} + \boldsymbol{A}$. In this way, the information from $\mathbf{u}$ to $\mathbf{x}$ is restricted to flow as if we were unrolling the causal model [34] and, as a result, the output of the $l-1$-th layer of the causal NF is given by:[3]

$$z_i^{l-1} = \tau_i(z_i^l; \mathbf{h}_i^{l-1}), \quad \text{where} \quad \mathbf{h}_i^{l-1} = c_i(\mathbf{z}_{\mathrm{pa}_i}^l). \tag{5}$$

Note that, by restricting each layer such that $\nabla_{\mathbf{z}^l}\boldsymbol{\tau}(\mathbf{z}^l) \equiv \boldsymbol{I} + \boldsymbol{A}$, there cannot exist shortcuts at the optima. For example, in Fig. 4, the (indirect) information of $u_1$ to generate $x_3$ by first generating

---

[3]The order has been reversed w.r.t. the causal NF definition from Eq. 3.

$x_2$ needs to go through the middle nodes in Fig. 4b. However, as shown by Sánchez-Martin et al. [34], we need at least $L = \mathrm{diam}(\boldsymbol{A})$ layers in our causal NF to avoid shortcuts and thus differentiate between the different direct and indirect causal paths connecting a pair of observed variables.

In contrast, if we only know the causal ordering, then the causal NF will need to rule out the spurious correlations by learning during training the necessary zeroes to fulfil causal consistency (Cor. 2), i.e., such that $\nabla_{\mathbf{x}} T_{\boldsymbol{\theta}}(\mathbf{x}) \equiv \boldsymbol{I} + \boldsymbol{A}$ and $\nabla_{\mathbf{u}} T_{\boldsymbol{\theta}}^{-1}(\mathbf{u}) \equiv \boldsymbol{I} + \sum_{n=1}^{\mathrm{diam}(\boldsymbol{A})} \boldsymbol{A}^n$.

**Abductive model**   Reminiscent to the abduction step [30], another natural choice is to model the inverse equations of the SCM as in Fig. 4d, hence building a causal NF from $\mathbf{x}$ to $\mathbf{u}$. Under a known causal graph, we can again use extra masking in the causal NF to force each layer $l$ to be structurally equivalent to $\boldsymbol{I} - \boldsymbol{A}$, such that

$$z_i^l = \tau_i(z_i^{l-1}; \mathbf{h}_i^l), \quad \text{where} \quad \mathbf{h}_i^l = c_i(\mathbf{z}_{\mathrm{pa}_i}^{l-1}). \tag{6}$$

Remarkably, this architecture is capable of capturing all indirect dependencies of $\mathbf{u}$ on $\mathbf{x}$, even with a single layer. This is a result of the autoregressive nature of the ANFs used here to build causal NFs, as they compute the inverse sequentially. In the example of Fig. 4, the indirect influence of $u_1$ on $x_3$ via $x_2$ has to necessarily generate $x_2$ first (Fig. 4a). Similar to the previous architecture, in the absence of a causal graph (i.e., when only the causal ordering is known), the causal NF will need to rely on optimization to discard all spurious correlations.

## 4.1   Necessary conditions

We next analyse the necessary conditions for the design of a causal NF to be able to accurately approximate and manipulate an SCM. A summary of the analysis can be found in Tab 1.

**Expressiveness**   The least restrictive condition is that the causal NF should be able to reach the optima and, as mentioned in §3, a single ANF layer (Eq. 3) is a universal TMI approximator [25].

**Identifiability**   In order to perform interventions as we describe later in §5, we need the causal NF to isolate the exogenous variables, so that we can associate them with their respective endogenous variables. As we saw in §3, if the causal NF is expressive enough, and *if it follows a valid causal ordering w.r.t. the true causal graph* $\boldsymbol{A}$, then Thm. 1 ensures that we can isolate the exogenous variables up to elementwise transformations.

**Causal consistency**   As stated in Cor. 2, the causal NF needs to share the causal dependencies of the SCM at the optima, meaning that their Jacobian matrices need to be structurally equivalent, i.e., $\nabla_{\mathbf{x}} T_{\boldsymbol{\theta}}(\mathbf{x}) \equiv \boldsymbol{I} - \boldsymbol{A}$ (Fig. 4d), and $\nabla_{\mathbf{u}} T_{\boldsymbol{\theta}}^{-1}(\mathbf{u}) \equiv \sum_{n=1}^{\mathrm{diam}(\boldsymbol{A})} \boldsymbol{A}^n + \boldsymbol{I}$ (Fig. 4c). Given the (partial) causal graph $\boldsymbol{A}$, the generative model in Eq. 5 by design holds $\nabla_{\mathbf{u}} T_{\boldsymbol{\theta}}^{-1}(\mathbf{u}) \equiv \sum_{n=1}^{\mathrm{diam}(\boldsymbol{A})} \boldsymbol{A}^n + \boldsymbol{I}$ for any sufficient number of layers $L \geq \mathrm{diam}\,\boldsymbol{A}$ (see [34, Prop. 1]), however, there might still exist spurious paths from $\mathbf{x}$ to $\mathbf{u}$. Similarly, the abductive model in Eq. 6, while may not remove all spurious paths from $\mathbf{x}$ to $\mathbf{u}$ if $L > 1$, ensures causal consistency when $L = 1$. In cases where the selected architecture for the causal NF does not ensure causal consistency by design, but we have access to the causal graph, we can use this extra information to regularize our MLE problem as

$$\underset{\boldsymbol{\theta}}{\mathrm{minimize}} \quad \mathrm{E}_{\mathbf{x}}\left[ -\log p(T_{\boldsymbol{\theta}}(\mathbf{x})) + \|\nabla_{\mathbf{x}} T_{\boldsymbol{\theta}}(\mathbf{x}) \odot (\mathbf{1} - \boldsymbol{A})\|_2 \right], \tag{7}$$

where $\mathbf{1}$ is a matrix of ones, thus penalizing spurious correlations from $\mathbf{x}$ to $T_{\boldsymbol{\theta}}(\mathbf{x})$.

Tab 1 summarizes the discussed properties of the considered design choices. Remarkably, the abductive model with a single layer (similar to Fig. 4d) *enjoys all the necessary properties of a causal NF by design*. That is, the abductive model with $L = 1$ is expressive, and causally consistent w.r.t. the provided causal graph $\boldsymbol{A}$, greatly simplifying the optimization process.

**Remark.** It is not straightforward why abductive models can be causally consistent by design, but generative models cannot. The answer lies on the structure of the problem. Intuitively, this is a consequence of the mapping $\mathbf{x} \to \mathbf{u}$ being structurally sparser ($u_i$ depends on $\mathrm{pa}_i$, see Fig. 4d) than that of $\mathbf{u} \to \mathbf{x}$ ($x_i$ depends on $\mathrm{an}_i$, see Fig. 4c). Therefore, ensuring causal consistency from $\mathbf{u}$ to $\mathbf{x}$ does not necessarily imply causal consistency from $\mathbf{x}$ to $\mathbf{u}$.

Table 1: Summary of the considered design choices, their induced properties, and their time complexity for density evaluation and sampling. Generative models design their forward pass as $\mathbf{u} \to \mathbf{x}$, and abductive models as $\mathbf{x} \to \mathbf{u}$. See §4 for an in-depth discussion.

| Design Choices | | Model Properties | | Time Complexity | |
| --- | --- | --- | --- | --- | --- |
| Network Type | Causal Asumption | Causal Consistency | | Sampling | Evaluation |
| | | $\mathbf{u} \to \mathbf{x}$ | $\mathbf{x} \to \mathbf{u}$ | | |
| $\mathbf{u} \to \mathbf{x}$ {  Generative | Ordering | ✗ | ✗ | $\mathcal{O}(L)$ | $\mathcal{O}(dL)$ |
| Generative | Graph $\mathbf{A}$ | ✓ | ✗ | $\mathcal{O}(L)$ | $\mathcal{O}(dL)$ |
| Abductive | Ordering | ✗ | ✗ | $\mathcal{O}(dL)$ | $\mathcal{O}(L)$ |
| $\mathbf{x} \to \mathbf{u}$ {  Abductive ($L > 1$) | Graph $\mathbf{A}$ | ✗ | ✗ | $\mathcal{O}(dL)$ | $\mathcal{O}(L)$ |
| Abductive ($L = 1$) | Graph $\mathbf{A}$ | ✓ | ✓ | $\mathcal{O}(dL)$ | $\mathcal{O}(L)$ |

## 5 Do-operator: enabling interventions and counterfactuals

In this section, we propose an implementation of the *do-operator* well-suited for causal NFs, such that we can evaluate the effect of interventions and counterfactuals [28]. The *do-operator* [30], denoted as $do(\mathrm{x}_i = \alpha)$, is a mathematical operator that simulates a physical intervention on an SCM $\mathcal{M}$, inducing an alternative model $\mathcal{M}^{\mathcal{I}}$ that fixes the observational value $\mathrm{x}_i = \alpha$, and thus removes any causal dependency on $\mathrm{x}_i$. Usually, the do-operator is implemented by yielding an SCM $\mathcal{M}^{\mathcal{I}} = (\tilde{\mathbf{f}}^{\mathcal{I}}, P_{\mathbf{u}})$ result of replacing the $i$-th component of $\tilde{\mathbf{f}}$ with a constant function, $\tilde{f}_i^{\mathcal{I}} := \alpha$. Unfortunately, this implementation of the do-operator only works for the recursive representation of the SCM (Fig. 4a), thus not generalizing to the different architecture designs of causal NFs discussed in §4 the previous section.

We instead propose to manipulate the SCM by modifying the exogenous distribution $P_{\mathbf{u}}$, while keeping the causal equations $\tilde{\mathbf{f}}$ untouched. Specifically, an intervention $do(\mathrm{x}_i = \alpha)$ updates $P_{\mathbf{u}}$, restricting the set of plausible $\mathbf{u}$ to those that yield the intervened value $\alpha$. We define the intervened SCM as $\mathcal{M}^{\mathcal{I}} = (\tilde{\mathbf{f}}, P_{\mathbf{u}}^{\mathcal{I}})$, where the density of $P_{\mathbf{u}}^{\mathcal{I}}$ is of the form

$$p^{\mathcal{I}}(\mathbf{u}) = \delta\left(\left\{\tilde{f}_i(\mathbf{x}_{\mathrm{pa}_i}, \mathrm{u}_i) = \alpha\right\}\right) \cdot \prod_{j \neq i} p_j(\mathbf{u}_j), \tag{8}$$

and where $\delta$ is the Dirac delta located at the unique value of $\mathrm{u}_i$ that yields $\mathrm{x}_i = \alpha$ after applying the causal mechanism $\tilde{f}_i$. This approach resembles the one proposed for soft interventions [10] and backtracking counterfactuals [37]. Moreover, as shown in App. C, it can be generalized for non-bijective causal equations. Note also that Eq. 8 is well-defined only if the set $\{\mathbf{u} \sim P_{\mathbf{u}} \,|\, \tilde{\mathbf{f}}_i(\boldsymbol{x}, \mathbf{u}) = \alpha\}$ is non-empty, i.e., if the intervened variable takes a *plausible* value (i.e., with positive density in the original causal model). Remarkably, this implementation works directly on the distribution of the exogenous variables and can be applied to any SCM representation (see Fig. 4), and therefore any of the architectures for the causal NF in §4.

**Implementation details** We take advantage of the autoregressive nature of causal NFs, and generate samples from Eq. 8 by: i) obtaining the exogenous variables $\mathbf{u} := T_{\boldsymbol{\theta}}(\mathbf{x})$; ii) replacing the observational value by its intervened value, $\mathrm{x}_i := \alpha$; and iii) by setting $\mathrm{u}_i$ to the value of the $i$-th component of $T_{\boldsymbol{\theta}}(\mathbf{x})$. If the causal NF has successfully isolated the exogenous variables (§3), and it preserves the true causal paths (§4), then the causal NF ensures that $\mathrm{x}_i = \alpha$ independently of the value of its ancestors, since $\mathrm{u}_i$ can be seen in Eq. 8 as a deterministic function of the given $\alpha$ and the value of its parents (and therefore its ancestors). We provide further details and the step-by-step algorithms to compute interventions and counterfactuals with causal NFs in App. C.

## 6 Empirical evaluation

In this section, we empirically validate the insights from §4, and compare causal NFs with previous works. Additional results and in-depth descriptions can be found in App. D.

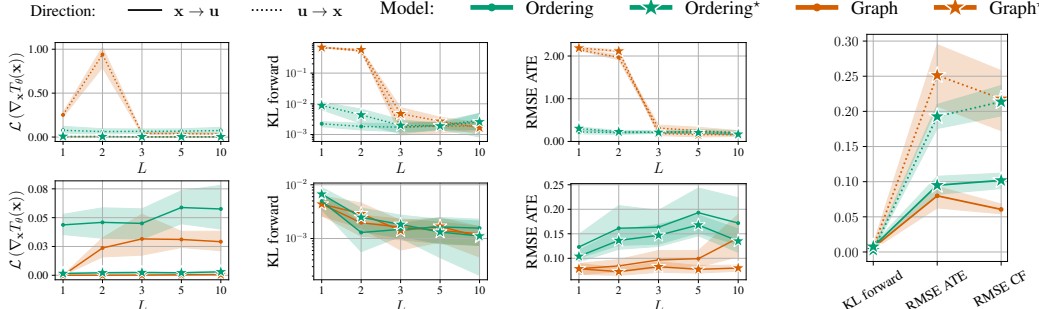

(a) Ablation study on a 4-node chain SCM, as we vary the number of layers $L$. Top: generative models ($\mathbf{u} \to \mathbf{x}$). Bottom: abductive models ($\mathbf{x} \to \mathbf{u}$).

(b) Comparison of the two best models of each row.

Figure 5: Ablation of different choices of the causal NF to be causally consistent, and capture the observational and interventional distributions. The use of regularization on the Jacobian (Eq. 7) is indicated with the $\star$ superscript. The abductive causal NF with information on $\boldsymbol{A}$ and $L = 1$ outperforms the rest of models across all metrics, demonstrating its efficacy and simplicity.

## 6.1 Ablation study

**Experimental setup** We evaluate every network combination described in Tab 1 on a 4-chain SCM (which has diameter 3 and a very sparse Jacobian) and assess the extent to which these models: i) capture the observational distribution, using $\mathrm{KL}(p_{\mathcal{M}} \| p_{\boldsymbol{\theta}})$; ii) remain causally consistent w.r.t. the original SCM, measured via $\mathcal{L}(\nabla_{\mathbf{x}} T_{\boldsymbol{\theta}}(\mathbf{x})) := \|\nabla_{\mathbf{x}} T_{\boldsymbol{\theta}}(\mathbf{x}) \cdot (\mathbf{1} - \boldsymbol{A})\|_2$ from Eq. 7; and iii) perform at interventional tasks, such as estimating the Average Treatment Effect (ATE) [29] and computing counterfactuals, both of which we measure with the RMSE w.r.t. the original SCM. Every experiment is repeated 5 times, and every causal NF uses Mask Autoregressive Flows (MAFs) [24] as layers.

**Results** Fig. 5 shows the result for different design choices of causal NFs. Specifically, we show: network design (generative $\mathbf{u} \to \mathbf{x}$ vs. abductive $\mathbf{x} \to \mathbf{u}$), causal knowledge (ordering vs. graph), number of layers $L$, and whether to use MLE with regularization (Eq. 4 vs. 7).

First, we see in Fig. 5a (top) that, as expected, the generative models ($\mathbf{u} \to \mathbf{x}$) using the causal graph cannot capture the SCM with $L < \mathrm{diam}\, \boldsymbol{A} = 3$. Furthermore, we observe that abductive models $\mathbf{x} \to \mathbf{u}$ (Fig. 5a, bottom) accurately fit the observational distribution, and that embedding the causal graph in the architecture significantly improves the ATE estimation.

Second, we now compare the two network designs in Fig. 5b, and observe that in general abductive models results in more accurate estimates of the observational distribution, as well as of interventional and counterfactual queries. Finally, we observe that regularization works well in all cases, yet it renders useless for the abductive model with $L = 1$ and knowledge on the graph, since it is causally consistent by design. In summary, our experiments confirm that, despite its simplicity, the causal abductive model with $L = 1$ outperforms the rest of design choices. As a consequence, in the following sections we will stick to this particular design choice, and refer to it as causal NF.

## 6.2 Non-linear SCMs

**Experimental setup** We compare our causal NF (causal, abductive, and with $L = 1$) with two relevant works: i) CAREFL [16], an abductive NF with knowledge on the causal ordering and affine layers; and ii) VACA [34], a variational auto-encoding GNN with knowledge on the graph. For fair comparison, every model uses the same budget for hyperparameter tuning, our causal NF uses affine layers, and CAREFL has been modified to use the proposed do-operator from §5 (as the original implementation only works in root nodes). We increase the complexity of the SCMs and consider: i) TRIANGLE, a 3-node SCM with a dense causal graph; ii) LARGEBD [11], a 9-node SCM with non-Gaussian $P_{\mathbf{u}}$ and made out of two chains with common initial and final nodes; and iii) SIMPSON [11], a 4-node SCM simulating a Simpson's paradox [33], where the relation between two variables changes if the SCM is not properly approximated.

**Results** The results are summarized in Tab 2. In a nutshell: *the proposed causal NF outperforms both CAREFL and VACA in terms of performance and computational efficiency.* VACA shows poor performance, and is considerably slower due to the complexity of GNNs. Our causal NF outperforms

Table 2: Comparison, on three non-linear SCMs, of the proposed causal NF, VACA [34], and CAREFL [16] with the do-operator proposed in §5. Results averaged over five runs.

| | | Performance | | | Time Evaluation (µs) | | |
|---|---|---|---|---|---|---|---|
| Dataset | Model | KL | $\text{ATE}_{\text{RMSE}}$ | $\text{CF}_{\text{RMSE}}$ | Training | Evaluation | Sampling |
| TRIANGLE NLIN [34] | Causal NF | $\mathbf{0.00_{0.00}}$ | $\mathbf{0.12_{0.03}}$ | $\mathbf{0.13_{0.02}}$ | $\mathbf{0.52_{0.07}}$ | $\mathbf{0.58_{0.07}}$ | $\mathbf{1.07_{0.12}}$ |
| | CAREFL$^\dagger$ | $\mathbf{0.00_{0.00}}$ | $\mathbf{0.12_{0.03}}$ | $0.17_{0.03}$ | $\mathbf{0.57_{0.18}}$ | $0.83_{0.26}$ | $1.68_{0.62}$ |
| | VACA | $7.71_{0.60}$ | $4.78_{0.01}$ | $4.19_{0.04}$ | $28.82_{1.21}$ | $23.00_{0.55}$ | $70.65_{3.70}$ |
| LARGEBD NLIN [11] | Causal NF | $\mathbf{1.51_{0.04}}$ | $\mathbf{0.02_{0.00}}$ | $\mathbf{0.01_{0.00}}$ | $\mathbf{0.52_{0.10}}$ | $\mathbf{0.60_{0.17}}$ | $\mathbf{3.05_{0.66}}$ |
| | CAREFL$^\dagger$ | $1.51_{0.05}$ | $0.05_{0.01}$ | $0.08_{0.01}$ | $0.84_{0.47}$ | $1.18_{0.17}$ | $8.25_{1.29}$ |
| | VACA | $53.66_{2.07}$ | $0.39_{0.00}$ | $0.82_{0.02}$ | $164.92_{11.10}$ | $137.88_{15.72}$ | $167.94_{25.75}$ |
| SIMPSON SYMPROD [11] | Causal NF | $\mathbf{0.00_{0.00}}$ | $\mathbf{0.07_{0.01}}$ | $\mathbf{0.12_{0.02}}$ | $0.59_{0.17}$ | $\mathbf{0.60_{0.11}}$ | $\mathbf{1.51_{0.30}}$ |
| | CAREFL$^\dagger$ | $\mathbf{0.00_{0.00}}$ | $0.10_{0.02}$ | $0.17_{0.04}$ | $\mathbf{0.49_{0.15}}$ | $0.81_{0.19}$ | $1.91_{0.33}$ |
| | VACA | $13.85_{0.64}$ | $0.89_{0.00}$ | $1.50_{0.04}$ | $49.26_{4.09}$ | $37.78_{3.41}$ | $79.20_{14.60}$ |

Table 3: Accuracy, F1-score, and counterfactual unfairness of the audited classifiers. Causal NFs enable both fair classifiers and accurate unfairness metrics. Results are averaged on five runs.

| | Logistic classifier | | | | SVM classifier | | | |
|---|---|---|---|---|---|---|---|---|
| | full | unaware | fair $\mathbf{x}$ | fair $\mathbf{u}$ | full | unaware | fair $\mathbf{x}$ | fair $\mathbf{u}$ |
| f1 | $72.28_{6.16}$ | $72.37_{4.90}$ | $59.66_{8.57}$ | $73.08_{4.38}$ | $76.04_{2.86}$ | $76.80_{5.82}$ | $68.28_{5.74}$ | $77.39_{1.52}$ |
| accuracy | $67.00_{3.83}$ | $66.75_{2.63}$ | $54.75_{5.91}$ | $66.50_{3.70}$ | $69.50_{3.11}$ | $71.00_{3.83}$ | $59.25_{2.99}$ | $69.75_{1.26}$ |
| unfairness | $5.84_{2.93}$ | $2.81_{0.72}$ | $0.00_{0.00}$ | $0.00_{0.00}$ | $6.65_{2.45}$ | $2.78_{0.40}$ | $0.00_{0.00}$ | $0.00_{0.00}$ |

CAREFL in counterfactual estimation tasks with identical observational fitting, showing once more the importance of being causally consistent. Even more, our causal NF is also quicker than CAREFL, as best-performing CAREFL architectures have in general more than one layer.

## 7 Use-case: fairness auditing and classification

To show the potential practical impact of our work, we follow the fairness use-case of Sánchez-Martin et al. [34] on the German Credit dataset [8]—a dataset from the UCI repository where the likelihood of individuals repaying a loan is predicted based on a small set of features, including sensitive attributes such as their sex. Extra details and results appear in App. E.

**Experimental setup**  As proposed by Chiappa [3], we use a partial graph which groups the 7 discrete features of the dataset in 4 different blocks with known causal relationships, putting in practice the results from §3.1. For the causal NF, we use the abductive model with a single non-affine neural spline layer [9]. Our ultimate goal is to train a causal NF that captures well the underlying SCM, and use it to train and evaluate classifiers that predict the (additional) binary feature *credit risk*, while remaining counterfactually fair w.r.t. the binary variable *sex*, $\mathrm{x}_S$.

In this setting, we call a binary classifier $\kappa : \mathbb{X} \to \{0, 1\}$ counterfactually fair [21] if, for all possible factual values $\mathbf{x}^{\mathrm{f}} \in \mathbb{X}$, the counterfactual unfairness remains zero. That is, if we have that $\mathrm{E}_{\mathbf{x}^{\mathrm{f}}}\left[P(\kappa(\mathbf{x}^{\mathrm{cf}}) = 1 \,|\, do(\mathrm{x}_S = 1), \mathbf{x}^{\mathrm{f}}) - P(\kappa(\mathbf{x}^{\mathrm{cf}}) = 1 \,|\, do(\mathrm{x}_S = 0), \mathbf{x}^{\mathrm{f}})\right] = 0$, where $\mathbf{x}^{\mathrm{cf}}$ is a counterfactual sample coming from the distribution $P(\mathbf{x}^{\mathrm{cf}} \,|\, do(\mathrm{x}_S = s), \mathbf{x}^{\mathrm{f}})$, for $s = 0, 1$.

Following Sánchez-Martin et al. [34], we audit: a model that takes all observed variables (*full*); an *unaware* model that leaves the sensitive attribute $\mathrm{x}_S$ out; a fair model that only considers non-descendant variables of $\mathrm{x}_S$ (*fair* $\mathbf{x}$); and, to demonstrate the ability to learn a counterfactually fair classifier, we include a classifier that takes $\mathbf{u} = T_{\boldsymbol{\theta}}(\mathbf{x})$ as input, but leaves $\mathrm{u}_S$ out (*fair* $\mathbf{u}$).

**Results**  Tab 3 summarizes the performance and unfairness of the classifiers, using logistic regression [6] and SVMs [5]. Here, we observe that by taking the non-sensitive exogenous variables from the causal NF, the obtained classifiers achieve comparable or better accuracy than the rest of the classifiers, while at the same time being counterfactually fair. Moreover, the estimations of unfairness obtained with the causal NF match our expectations [21], with *full* being the most unfair, followed by *aware* and the two fair models. With this use-case, we demonstrate that *Causal NFs may indeed be a valuable asset for real-world causal inference problems*.

# 8 Concluding remarks

In this work, we have shown—both theoretically and empirically—that causal NFs are a natural choice to learn a broad class of causal data-generating processes in a principled way. Specifically, we have proven that causal NFs can match the observational distribution of an underlying SCM, and that in doing so the ANF needs to be causally consistent. However, as limited data availability and local optima may hamper reaching these solutions in practice, we have explored different network designs, exploiting the available knowledge on the causal graph. Moreover, we have provided causal NFs with a do-operator to efficiently solve causal inference tasks. Finally, we have empirically validated our findings, and demonstrated that our causal NF framework: i) outperforms competing methods; and ii) can deal with mixed data and partial knowledge on the causal graph.

**Practical limitations**    Despite considering a broad class of SCMs, we have made several assumptions that, while being standard, may not hold in some application scenarios. With regard to our causal assumptions, the presence of unmeasured hidden confounders may break our causal sufficiency assumption; mismatches between the true causal graph (e.g., it may contain cycles) and our assumed graph/ordering may lead to poor estimates of interventional and counterfactual queries; and the non-bijective true causal dependencies may invalidate our theoretical and thus practical findings. Besides, we have focused on MLE estimation for learning the causal NF. However, MLE does not test the independency of the exogenous variables during training, which would also break our causal sufficiency assumption.

**Future work**    We firmly believe that our work opens a number of interesting directions to explore. Naturally, we would like to address current limitations by, e.g., using interventional data to address the existence of hidden confounders [14, 23], explore alternative losses other than MLE (e.g., flow matching [22]). Moreover, it would be exciting to see causal NFs applied to other problems such as causal discovery [11], fair decision-making [21], or neuroimaging [12], among others. However, we would like to stress that, in the above contexts, it would be essential to validate the suitability of our framework (e.g., using experimental data) to prevent potential harms.

# Acknowledgements

We would like to thank Batuhan Koyuncu and Jonas Klesen for their invaluable feedback, as well as the anonymous reviewers whose feedback helped us improve this manuscript. This project is funded by DFG grant 389792660 as part of TRR 248 – CPEC. Pablo Sánchez Martín thanks the German Research Foundation through the Cluster of Excellence "Machine Learning – New Perspectives for Science", EXC 2064/1, project number 390727645 for generous funding support. The authors also thank the International Max Planck Research School for Intelligent Systems (IMPRS-IS) for supporting Pablo Sánchez Martín.

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

# Appendix

## Table of Contents

# A  Theory of causal normalizing flows

## A.1  Identifiability results

First, we provide a more detailed explanation on the connection between the results from §3 and those from the work of Xi and Bloem-Reddy [38]. We consider it important to clarify that the definition of identifiability that we use is the same as [38, Def. 2]. Specifically, this definition is one better suited for deep learning models, which is concerned with recovering the variables $\mathbf{u}$ and *one* parametrization that perfectly matches the original generator. In other words, with this definition we aim to recover one parametrization of a neural network which provides the generator function, but not the *exact* parametrization of the generator that generated the data.

We also want to clarify that Thm. 1 from the main paper corresponds to [38, Prop. 5.2], which we rewrote (without changing its content) to plain English and to match our particular setting. We now provide the proof for Cor. 2:

**Corollary 2** (Causal consistency). If a causal NF $T_{\boldsymbol{\theta}}$ isolates the exogenous variables of an SCM $\mathcal{M}$, then $\nabla_{\mathbf{x}} T_{\boldsymbol{\theta}}(\mathbf{x}) \equiv \boldsymbol{I} - \boldsymbol{A}$ and $\nabla_{\mathbf{u}} T_{\boldsymbol{\theta}}^{-1}(\mathbf{u}) \equiv \boldsymbol{I} + \sum_{n=1}^{\mathrm{diam}(\boldsymbol{A})} \boldsymbol{A}^n$, where $\boldsymbol{A}$ is the causal adjacency matrix of $\mathcal{M}$. In other words, $T_{\boldsymbol{\theta}}$ is causally consistent with the true data-generating process, $\mathcal{M}$.

**Proof.** Assume that we have a flow $T_{\boldsymbol{\theta}}$ that does indeed isolate the exogenous variables, meaning that the $i$-th output of the flow, $T_{\boldsymbol{\theta}}(\mathbf{x})_i$, is related with the true exogenous variable, $\mathrm{u}_i$, by an invertible function that only depends on it.

As explained in §3, this means that for a variable $\mathbf{u} \sim P_{\mathcal{M}}$, we have that $T_{\boldsymbol{\theta}}(\mathbf{f}(\mathbf{u})) \sim P_{\boldsymbol{\theta}}$ and $T_{\boldsymbol{\theta}}(\mathbf{f}(\mathbf{u})) = \boldsymbol{h}(\mathbf{u}) = (h_1(\mathrm{u}_1), h_2(\mathrm{u}_2), \ldots, h_d(\mathrm{u}_d))$.

But we know the true generator, whose $i$-th exogenous variable is given by $\mathrm{u}_i = f_i^{-1}(\mathbf{x}_{\mathrm{pa}_i}, \mathrm{x}_i)$ (the inverse of $f_i$ w.r.t $\mathrm{u}_i$) and, putting all together,

$$T_{\boldsymbol{\theta}}(\mathbf{x})_i = h_i(\mathrm{u}_i) = h_i(f_i^{-1}(\mathbf{x}_{\mathrm{pa}_i}, \mathrm{x}_i)), \tag{9}$$

which is a function of *only* the parents of $\mathrm{x}_i$ and $\mathrm{x}_i$ itself.

If we call $\mathbf{u} = \mathbf{f}^{-1}(\mathbf{x}) := (f_1^{-1}(\mathbf{x}_{\mathrm{pa}_1}, \mathrm{x}_1), f_2^{-1}(\mathbf{x}_{\mathrm{pa}_2}, \mathrm{x}_2), \ldots, f_d^{-1}(\mathbf{x}_{\mathrm{pa}_d}, \mathrm{x}_d))$ the inverse of the SCM $\mathcal{M}$ that writes $\mathbf{u}$ as a function of $\mathbf{x}$ (see App. B for an example), then it is clear that

$$\nabla_{\mathbf{x}} T_{\boldsymbol{\theta}}(\mathbf{x}) = \nabla_{\mathbf{x}}(\boldsymbol{h} \circ \mathbf{f}^{-1})(\mathbf{x}) = \nabla_{\mathbf{u}} \boldsymbol{h}(\mathbf{u}) \cdot \nabla_{\mathbf{x}} \mathbf{f}^{-1}(\mathbf{x}) = \boldsymbol{D} \cdot \nabla_{\mathbf{x}} \mathbf{f}^{-1}(\mathbf{x}) \equiv \boldsymbol{I} - \boldsymbol{A}, \tag{10}$$

where $\boldsymbol{D}$ is a diagonal matrix and $\boldsymbol{A}$ the adjacency matrix of the causal graph induced by $\mathcal{M}$.

Similarly, $T_{\boldsymbol{\theta}}^{-1}(\boldsymbol{h}(\mathbf{u})) = \mathbf{x} = \mathbf{f}(\mathbf{u})$ and $T_{\boldsymbol{\theta}}^{-1}(\boldsymbol{h}(\mathbf{u}))_i = \mathrm{x}_i = f_i(\mathbf{u}_{\mathrm{an}_i}, \mathrm{u}_i)$, which again implies that:

$$\nabla_{\tilde{\mathbf{u}}} T_{\boldsymbol{\theta}}^{-1}(\tilde{\mathbf{u}}) \equiv (\boldsymbol{I} - \boldsymbol{A})^{-1} = \boldsymbol{I} + \sum_{n=1}^{\mathrm{diam}(\boldsymbol{A})} \boldsymbol{A}^n, \tag{11}$$

where we call $\tilde{\mathbf{u}} = \boldsymbol{h}(\mathbf{u})$ the variable $\mathbf{u}$ recovered by $T_{\boldsymbol{\theta}}$, and where we have omitted $\boldsymbol{h}$ as its Jacobian matrix is diagonal. Note that the infinite sum above vanishes at $n = \mathrm{diam}\,\boldsymbol{A}$ since $\boldsymbol{A}$ is triangular with diagonal zero. Q.E.D.

## A.2  Extension to real-world settings

### A.2.1  Discrete data

In this section, we describe how to extend the results presented in the main text for the case where one observed variable, $\mathrm{x}_i$, is discrete. To this end, we restate the more general data-generative process assumed by Xi and Bloem-Reddy [38], which we used for the theoretical part of the manuscript.

Following the notation of the manuscript, say that we have a data-generating process without recursions, that is, we have a function $\mathbf{f}$ that maps $\mathbf{u}$ to $\mathbf{x}$. Let us assume, without loss of generality, that only the $i$-th observed variable is discrete, and let us focus on the way this variable is generated, dropping the subindex $i$ along the way to avoid clutter. Now, Xi and Bloem-Reddy [38] additionally

consider the existence of a fixed noise distribution $P_\varepsilon$ and mechanism $g$, such that the observed variable $\mathrm{x}_i$ is generated as,

$$\mathbf{u} := (\mathrm{u}_1, \mathrm{u}_2, \ldots, \mathrm{u}_d) \sim P_\mathbf{u}\,, \qquad \varepsilon \sim P_\varepsilon\,, \qquad \mathrm{x}_i = g(f_i(\mathbf{u}_{\mathrm{an}_i}, \mathrm{u}_i), \varepsilon)\,, \qquad (12)$$

with $\varepsilon \perp\!\!\!\perp \mathbf{u}$, and where they study the noiseless case under the following assumption: if for two generative processes with $\varepsilon_a \stackrel{\mathrm{d}}{=} \varepsilon_b$, then $g(f_i(\mathbf{u}_a), \varepsilon_a) \stackrel{\mathrm{d}}{=} g(f_i(\mathbf{u}_b), \varepsilon_b)$ if and only if $f_i(\mathbf{u}_a) \stackrel{\mathrm{d}}{=} f_i(\mathbf{u}_b)$, where $\stackrel{\mathrm{d}}{=}$ denotes equal in distribution.

Just as we do with the rest of variables, we also make the assumption that the observed variable $\tilde{\mathrm{x}}_i$ is the transformation of a continuous exogenous variable, $\mathrm{u}_i$, with a function $\tilde{f}_i$ that fulfils our assumptions (i.e., that $\tilde{f}_i$ is a diffeomorphism) that has undergone a quantization process, i.e., $\mathrm{x}_i = f_i(\mathbf{u}_{\mathrm{an}_i}, \mathrm{u}_i) := \lfloor \tilde{f}_i(\mathbf{u}_{\mathrm{an}_i}, \mathrm{u}_i) \rfloor$. Therefore, it is clear that $f_i$ is no longer bijective, as we are clamping real numbers into integers, and that the observational distribution of $\mathrm{x}_i$ is discrete.

We take advantage of the noise assumption above, and dequantize the observed variable $\mathrm{x}_i$ by assuming an additive noise mechanism such that $\mathrm{x}_i := f_i(\mathbf{u}_{\mathrm{an}_i}, \mathrm{u}_i) + \varepsilon$, with $\varepsilon$ distributed between the unit interval with any continuous distribution (we take in our experiments $P_\varepsilon = \mathcal{U}(0,1)$). With this process: i) we have made $\tilde{\mathrm{x}}_i$ again a continuous random variable, as the sum of independents discrete and continuous random variables is a continuous random variable; and ii) the original distribution of the noiseless observed variables is always recoverable $P(\tilde{\mathrm{x}}_i = c) = P(c \le \mathrm{x}_i \le c+1)$.

More importantly, all the theoretical insights from the work of Xi and Bloem-Reddy [38] can still be used, working with the noisy case rather than the noiseless one. Indeed, as for their analysis they assume a single $\mathbf{u}$ in the domain of the generator, we can merge the generator and noise mechanism $g \circ f_i : \mathbb{R} \to \mathbb{R}$ (rather than $g \circ f_i : \mathbb{R} \times [0,1] \to \mathbb{R}$), by mapping the non-injective part of $\mathbf{u}$ to $\varepsilon$ itself, i.e., by using the function $(g \circ f_i)(\mathbf{u}_{\mathrm{an}_i}, \mathrm{u}_i) = \lfloor \tilde{f}_i(\mathbf{u}_{\mathrm{an}_i}, \mathrm{u}_i) \rfloor + F_\varepsilon^{-1}(\tilde{f}_i(\mathbf{u}_{\mathrm{an}_i}, \mathrm{u}_i) - \lfloor \tilde{f}_i(\mathbf{u}_{\mathrm{an}_i}, \mathrm{u}_i) \rfloor)$, where $F_\varepsilon^{-1}$ is the quantile function of $P_\varepsilon$. This new function is a diffeomorphism almost everywhere, as it is a composition of a.e. diffeomorphisms, and we have effectively replaced the noise variable by the floating part of $\tilde{f}_i(\mathbf{u}_{\mathrm{an}_i}, \mathrm{u}_i)$ before quantization. Moreover, note that if the noise is uniformly distributed, $P_\varepsilon = \mathcal{U}(0,1)$, we have that $g \circ f_i = \tilde{f}_i$ (if $\mathrm{x}_i$ were discretized by taking its integer part).

In short, by adding noise to discrete variables while keeping them recoverable, we can learn a mapping between continuous variables that learns a version of the generator function before the observed values were somehow discretized. Importantly, the observed discrete distribution is always recoverable, independently of whether we learn the (unknown and unrecoverable) underlying continuous distribution before being discretized.

### A.2.2 Partial knowledge

In this section, we explain how to expand our framework to settings in which we have partial information about the causal graph of $\mathcal{M}$. That is, we know the causal ordering $\pi$ (so we know that half of the causal relationships, the upper diagonal), and we are certain about some other causal relationships (edges on $\mathbf{A}$), but not all of them.

To this end, first let us first introduce the way we deal with partial knowledge, and then clarify the theoretical implications that it has with respect to the theory introduced in §5.

**The method** Similar to §4, let us motivate the method with an illustrative example. Suppose that we are given an SCM such as the one in Fig. 6a, where we know all relationships but the one between $\mathrm{x}_2$ and $\mathrm{x}_3$. Note that, in this case, we lack even information about the causal ordering. Indeed, there are three possible outcomes: i) the edge $\mathrm{x}_2 \to \mathrm{x}_3$ could exist (Fig. 6b); ii) the edge $\mathrm{x}_3 \to \mathrm{x}_2$ could exist (Fig. 6c); or iii) both could exist simultaneously (Fig. 6d), and hence there is a confounder between them. However, we do not know which of the three options is the correct one.

Let us switch now to Fig. 7. To solve the original problem (Fig. 7a, one natural approach is to group the nodes with unknown relationships—assuming that all unknown edges may exist—and maximize the observed likelihood (Fig. 7b). This, effectively, is equivalent to applying an ANF to the known relationships, and using a general-Jacobian NF to learn the joint of the block variables. However, if we want to keep using exclusively ANFs (Fig. 7c), we can learn the joint distribution within the blocks with an ANF using a fixed ordering (which it can always do, as it is a universal density approximator [25]). The only subtle detail here is that, in that case, we need to increase the

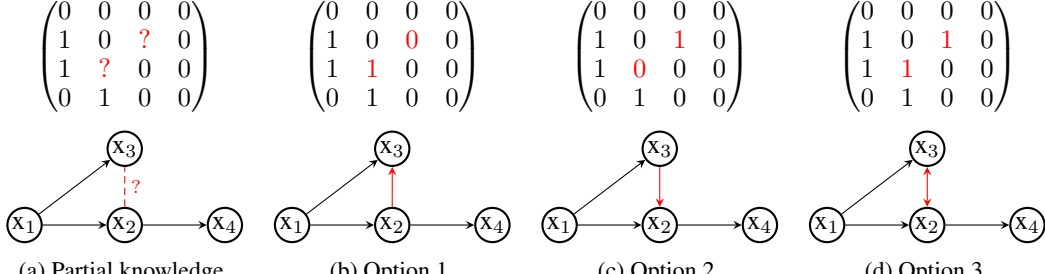

(a) Partial knowledge.      (b) Option 1.      (c) Option 2.      (d) Option 3.

Figure 6: Example of an SCM with partial knowledge about the causal graph (a) and possible outcomes: (b) in the actual SCM only the edge $x_2 \to x_3$ exists; (c) only the edge $x_3 \to x_2$ exists; (d) both edges exist (and therefore there exists a confounder between them).

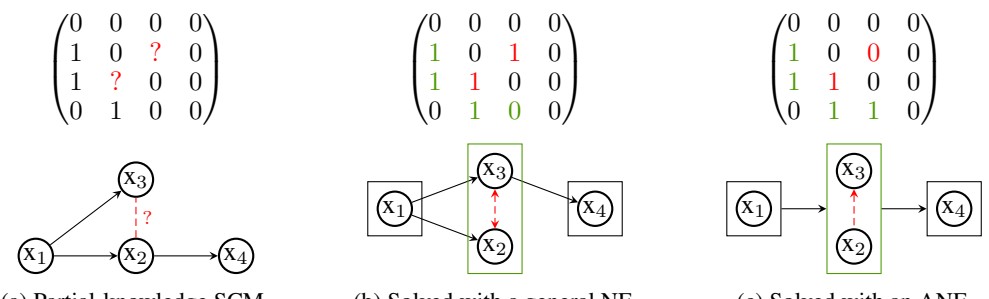

(a) Partial-knowledge SCM.      (b) Solved with a general NF.      (c) Solved with an ANF.

Figure 7: Illustrative example (same as in Fig. 6) applying our method for partial information. First, we apply Tarjan's algorithm [35] to find the SCCs of the graph (rectangles) and build a new DAG where each node is a subset of the original nodes. If, for the SCCs, we use an NF with a general Jacobian matrix (b), we keep the individual edges and treat each SCC as a block. If we instead use an ANF (c), we pick an arbitrary order within each SCC, and merge the individual edges into SCC-wide edges. Red represents intra-SCC edges, and green inter-SCC edges. See App. A.2.2 for more details.

granularity of all inter-block edges from node- to block-wise relationships, assuming that an edge exists if it exists for at least one of the elements of the block. To see why this is necessary, assume that the real graph of the example is Fig. 6c, yet we use an ANF with the ordering $\pi = (1\,2\,3\,4)$ and the graph $A$ without inter-block modifications (i.e., the adjacency matrix of Fig. 6b). In that case, we would have that $x_4$ depends on $x_3$ through $x_2$. However, a causally consistent NF w.r.t. $A$ would not be able to model that dependency, and thus $x_4$ would depend on $u_1$, $u_2$, and $u_4$ but not on $u_3$. With this approach, the ANF can model every case from Fig. 6, and it would need to remove the extra spurious relationships through optimization.

Therefore, to reuse our existing results from §5, the method that we have adopted is the one from Fig. 7c, which can be described in the following steps:

1. Run Tarjan's algorithm [35] to group all nodes by their SCCs (note that, unlike in the given example, there could be more than one cluster of unknown relationships).
2. Choose an ordering that is consistent with the known inter-SCC edges, fixing the edges within the SCCs.
3. Move from node-edges to SCC-edges. In practice, this means introducing edges between every pair of edges of two SCCs, if there exists at least one edge between them.
4. Solve the MLE problem with an ANF as in the main manuscript.

**The theory**   Very conveniently, the method described above fits almost-perfectly into the already-covered theory in §3. To see this, note that our identifiability theory and following results required a fixed ordering, but it does not need to be the causal one: we can always find an equivalent SCM following the selected ordering by applying KR transports as described in §3. Therefore, the technical

implications of both Thm. 1 and Cor. 2 still apply to this fixed ordering, we only need to re-state their implications with respect to the true causal data-generating process $\mathcal{M}$.

To this end, we only need to note that all possible graphs, once reduced into a DAG using the partition of SCCs (as in Fig. 7c), are exactly equal. In other words, every possible graph shares *the same causal dependencies between SCCs* with the other possible graphs. If we start treating them like block, calling $\{x_i\} \subset \boldsymbol{S}_i \subset \{1, 2, \ldots, d\}$ the SCC of the $i$-th node, and $\mathrm{pa}_{\boldsymbol{S}_i}$ and $\mathrm{an}_{\boldsymbol{S}_i}$ the parents and ancestors of every node in the SCC $\boldsymbol{S}_i$, then it is clear that we can write for every graph $\tilde{\boldsymbol{A}}$ its observed variables as $x_i = \tilde{f}_i(\mathbf{x}_{\mathrm{pa}_{\boldsymbol{S}_i}}, \tilde{\mathbf{u}}_{\boldsymbol{S}_i})$ and, more importantly, we can write its "exogenous" variables as a function of the true ones, i.e., $\tilde{\mathbf{u}}_{\boldsymbol{S}_i} = \mathrm{KR}(\mathbf{u}_{\boldsymbol{S}_i})$, where KR is the Knöthe-Rosenblatt transport [19, 31]. Note also that it does not depend on the data, i.e., $\tilde{\mathbf{u}}_{\boldsymbol{S}_i} \perp\!\!\!\perp \mathbf{x} | \mathbf{u}_{\boldsymbol{S}_i}$.

**Theorem 3** (Identifiability – Partial knowledge). If an element of $\mathcal{F} \times \mathcal{P}_{\mathbf{u}}$, and another from $\tilde{\mathcal{F}} \times \mathcal{P}_{\mathbf{u}}$, where $\mathcal{F}$ and $\tilde{\mathcal{F}}$ are TMI maps with different intra-SCC orders (see above), generate the same observational distribution, then the two processes differ by an invertible, SCC-wise transformation of the variables $\mathbf{u}$.

**Proof.** Call $\mathcal{M}$ and $\tilde{\mathcal{M}}$ the elements from $\mathcal{F} \times \mathcal{P}_{\mathbf{u}}$ and $\tilde{\mathcal{F}} \times \mathcal{P}_{\mathbf{u}}$, respectively.

W.l.o.g. pick $\mathcal{M}$, and apply a KR transport to write it down as another element of $\tilde{\mathcal{F}} \times \mathcal{P}_{\mathbf{u}}$, call it $\hat{\mathcal{M}}$, with identical observational distribution as both $\mathcal{M}$ and $\tilde{\mathcal{M}}$.

Using Thm. 1, we know that the elements of $\tilde{\mathcal{M}}$ and $\hat{\mathcal{M}}$ differ by an invertible, component-wise transformation $\boldsymbol{h}$. Moreover, we can write $\hat{\mathbf{u}}_{\boldsymbol{S}_i}$ as a function of $\mathbf{u}_{\boldsymbol{S}_i}$, as argued above, where $\{i\} \subset \boldsymbol{S}_i \subset \{1, 2, \ldots, d\}$ are the indexes of the SCC that contains $u_i$. Putting it all together:

$$\tilde{u}_i = h_i(\hat{u}_i) = h_i(\mathrm{KR}_i(\mathbf{u}_{\boldsymbol{S}_i})), \tag{13}$$

and, in vectorial form, for each SCC $\boldsymbol{S}_i$,

$$\tilde{\mathbf{u}}_{\boldsymbol{S}_i} = \boldsymbol{h}_{\boldsymbol{S}_i}(\mathrm{KR}_{\boldsymbol{S}_i}(\mathbf{u}_{\boldsymbol{S}_i})) = (\boldsymbol{h}_{\boldsymbol{S}_i} \circ \mathrm{KR}_{\boldsymbol{S}_i})(\mathbf{u}_{\boldsymbol{S}_i}) \tag{14}$$

Q.E.D.

**Corollary 4** (Causal consistency – Partial knowledge). If a causal NF $T_{\boldsymbol{\theta}}$, with partial knowledge of the causal graph, SCC-wise isolates the exogenous variables of an SCM $\mathcal{M}$, then $T_{\boldsymbol{\theta}}$ is causally consistent with the true data-generating process, $\mathcal{M}$, with respect to each SCC.

**Proof.** The proof is identical to the one for Cor. 2, but using arguments with respect to the reduced graph after grouping all nodes in their respective SCCs.

Specifically, we can write using Thm. 3 the output of the flow as a function of the exogenous variables, $T_{\boldsymbol{\theta}}(\mathbf{x})_i = \boldsymbol{h}(\mathbf{u}_{\boldsymbol{S}_i})$, and using the true causal generator, we have

$$T_{\boldsymbol{\theta}}(\mathbf{x})_i = \boldsymbol{h}(\mathbf{u}_{\boldsymbol{S}_i}) = \boldsymbol{h}(f_{\boldsymbol{S}_i}(\mathbf{x}_{\mathrm{pa}_{\boldsymbol{S}_i}}, \mathbf{x}_{\boldsymbol{S}_i})), \tag{15}$$

and hence the gradients agree with those from $\mathcal{M}$, *when looking at the reduced graph*. Q.E.D.

Thm. 3 and Cor. 4 provide analogues to those results from the main manuscript. It is important to note, however, that causal consistency refers to the causal relationships between SCCs *as a whole*, i.e., not at the causal relationships between individual nodes. The reason behind this is the same as why we had to introduced spurious edges in Fig. 7c: as we fix an ordering within each SCC, we may not be able to model indirect dependencies of nodes of one SCC to another unless we artificially introduce shortcuts. As such, *every result from the main paper holds*, if we treat each SCC (or block) as a whole. That is, when we reason about SCCs instead of individual nodes (note that if the whole causal graph is known every SCC contains a single node), we can safely talk about SCC-identifiability, causal SCC-consistency, and we can perform interventions and compute counterfactuals on SCCs.

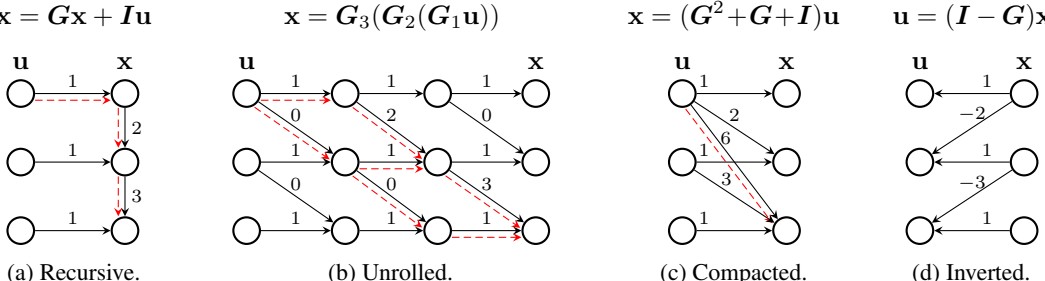

$$\mathbf{x} = \boldsymbol{G}\mathbf{x} + \boldsymbol{I}\mathbf{u} \qquad \mathbf{x} = \boldsymbol{G}_3(\boldsymbol{G}_2(\boldsymbol{G}_1\mathbf{u})) \qquad \mathbf{x} = (\boldsymbol{G}^2 + \boldsymbol{G} + \boldsymbol{I})\mathbf{u} \qquad \mathbf{u} = (\boldsymbol{I} - \boldsymbol{G})\mathbf{x}$$

(a) Recursive.   (b) Unrolled.   (c) Compacted.   (d) Inverted.

Figure 8: Example of the linear SCM $\{x_1 := u_1 \, ; \, x_2 := 2x_1 + u_2 \, ; \, x_3 := 3x_2 + u_3\}$ written (a) in its usual recursive formulation; (b) without recursions, with each step made explicit; (c) without recursions, as a single function; and (d) writing $\mathbf{u}$ as a function of $\mathbf{x}$. The red dashed arrows show the influence of $u_1$ on $x_3$ for all equations from $\mathbf{u}$ to $\mathbf{x}$, with the compacted version exhibiting shortcuts (see §4). Note that in the linear case we have $\boldsymbol{A} := \boldsymbol{G} \neq \boldsymbol{0}$, and that $\boldsymbol{G}_1, \boldsymbol{G}_2, \boldsymbol{G}_3 \preceq \boldsymbol{G} + \boldsymbol{I}$ are any three matrices such that their product equals $\boldsymbol{G}^2 + \boldsymbol{G} + \boldsymbol{I}$.

# B  The multiple representations of SCMs

## B.1  Illustrative example

In this section, we delve a bit deeper into the illustrative example from the main paper (§4), and write down the maths to understand how are these different models equivalent. For convenience to the reader, we have replicated Fig. 4 in the appendix as Fig. 8.

**Recursive SCM** As explained in the manuscript, the usual way of describing an SCM $\mathcal{M}$ is by providing its recursive equations. In our example (Fig. 8a), we have a linear SCM of the form

$$\begin{cases} x_1 = u_1 \\ x_2 = 2x_1 + u_2 \\ x_3 = 3x_2 + u_3 \end{cases}, \tag{16}$$

which we can compactly write as $\mathbf{x} = \boldsymbol{G}\mathbf{x} + \boldsymbol{I}\mathbf{u}$. However, the recursive equations are not the most convenient ones, as they entail solving the system iteratively according to its causal dependencies.

**Unrolled SCM** Instead, we can write the equations as a function from $\mathbf{u}$ to $\mathbf{x}$ directly. To do this, we can proceed and unroll the equations:

$$\begin{cases} x_1 = u_1 \\ x_2 = 2x_1 + u_2 \\ x_3 = 3x_2 + u_3 \end{cases} \Rightarrow \begin{cases} x_1 = u_1 \\ x_2 = 2u_1 + u_2 \\ x_3 = 3(2x_1 + u_2) + u_3 \end{cases} \Rightarrow \begin{cases} x_1 = u_1 \\ x_2 = 2u_1 + u_2 \\ x_3 = 3(2u_1 + u_2) + u_3 \end{cases}, \tag{17}$$

which we can write as a multi-step function:

$$\begin{cases} z_1^1 = u_1 \\ z_2^1 = u_2 \\ z_3^1 = u_3 \end{cases} \Rightarrow \begin{cases} z_1^2 = z_1^1 \\ z_2^2 = 2z_1^1 + z_2^1 \\ z_3^2 = z_3^1 \end{cases} \Rightarrow \begin{cases} x_1 = z_1^2 \\ x_2 = z_2^2 \\ x_3 = 3z_2^2 + z_3^2 \end{cases}, \tag{18}$$

that we can once again compactly write as a series of linear operations $\mathbf{x} = \boldsymbol{G}_3(\boldsymbol{G}_2(\boldsymbol{G}_1\mathbf{u}))$. Note that the matrices $\boldsymbol{G}_1, \boldsymbol{G}_2, \boldsymbol{G}_3$ are not unique, and that they are valid as long as they are at most as sparse as $\boldsymbol{G}$, and produce the same final output.

**Compacted SCM** Another natural step here is to compress this sequence of linear equations into a single operation. That is, use directly the linear operation described at the end of Eq. 17:

$$\begin{cases} x_1 = u_1 \\ x_2 = 2u_1 + u_2 \\ x_3 = 6u_1 + 3u_2 + u_3 \end{cases}. \tag{19}$$

And we can derive the same result in vectorial form:

$$\mathbf{x} = \boldsymbol{G}\mathbf{x} + \boldsymbol{I}\mathbf{u} \Rightarrow \mathbf{x} = \boldsymbol{G}(\boldsymbol{G}\mathbf{x} + \boldsymbol{I}\mathbf{u}) + \boldsymbol{I}\mathbf{u} \Rightarrow \mathbf{x} = \boldsymbol{G}(\boldsymbol{G}(\boldsymbol{G}\mathbf{x} + \boldsymbol{I}\mathbf{u}) + \boldsymbol{I}\mathbf{u}) + \boldsymbol{I}\mathbf{u} \Rightarrow$$

$$\mathbf{x} = \underset{0}{\boldsymbol{G}^3\mathbf{x}} + \boldsymbol{G}^2\mathbf{u} + \boldsymbol{G}\mathbf{u} + \boldsymbol{I}\mathbf{u} = (\boldsymbol{G}^2 + \boldsymbol{G} + \boldsymbol{I})\mathbf{u}\,.$$

Unfortunately, as described in §4, in this form we cannot longer distinguish indirect paths: we have collapsed all paths into direct paths. Even worse, if there were more than one path between two given nodes, we have combined their contributions into a single path, making it quite difficult to disentangle. This effect can be seen as analogous to the common process in cryptography for sharing secrets: if you have two primes (paths), it is fairly easy to multiply them and obtain their product, but if you have their product (collapsed paths), performing prime factorization of the number is prohibitive.

**Inverted SCM** Finally, we can take any of these different representations and invert the equations to go from $\mathbf{x}$ to $\mathbf{u}$. In this case, it is easier to work with the original equations:

$$\begin{cases} \mathbf{x}_1 = \mathbf{u}_1 \\ \mathbf{x}_2 = 2\mathbf{x}_1 + \mathbf{u}_2 \\ \mathbf{x}_3 = 3\mathbf{x}_2 + \mathbf{u}_3 \end{cases} \Rightarrow \begin{cases} \mathbf{u}_1 = \mathbf{x}_1 \\ \mathbf{u}_2 = \mathbf{x}_2 - 2\mathbf{x}_1 \\ \mathbf{u}_3 = \mathbf{x}_3 - 3\mathbf{x}_2 \end{cases}, \tag{20}$$

and, in vectorial form:

$$\mathbf{x} = \boldsymbol{G}\mathbf{x} + \boldsymbol{I}\mathbf{u} \Rightarrow \mathbf{u} = (\boldsymbol{I} - \boldsymbol{G})\mathbf{x}\,. \tag{21}$$

As discussed in the main manuscript, this turns out to be a really convenient SCM representation to work with, as we can obtain the exogenous variables in one go while being causally consistent.

## B.2 Non-linear SCM representations

In this section, we discuss how the same representation and reasoning about the causal relationships of a linear SCM from App. B.1 can be to the general case.

To this end, assume that we have a non-linear SCM $\mathcal{M}$ of the form $\mathbf{x} = \tilde{\mathbf{f}}(\mathbf{x}, \mathbf{u})$ where, to ease the reader, imagine that it has the same causal graph as the linear example, so that the reader can use Fig. 8 as a reference again, i.e., assume that

$$\left(\nabla_{\mathbf{x}}\tilde{\mathbf{f}}(\mathbf{x}, \mathbf{u}) \neq \mathbf{0}\right) = \begin{pmatrix} 0 & 0 & 0 \\ 1 & 0 & 0 \\ 0 & 1 & 0 \end{pmatrix} \quad \text{and} \quad \left(\nabla_{\mathbf{u}}\tilde{\mathbf{f}}(\mathbf{x}, \mathbf{u}) \neq \mathbf{0}\right) = \begin{pmatrix} 1 & 0 & 0 \\ 0 & 1 & 0 \\ 0 & 0 & 1 \end{pmatrix}, \tag{22}$$

where $\mathbf{0}$ is the constant zero function with the same domain and codomain as the Jacobian matrices.

**Recursive SCM** Already extensively discussed. This is the representation an SCM is given as.

**Unrolled SCM** Just as before, we can unroll the equations by having multiple functions $\mathbf{z}^l = \mathbf{f}_l(\mathbf{z}^{l-1})$, and in each one we unroll those equations for which we already know the non-recursive equation of its parents, leaving all the other fixed (identity functions). As before, we can write these multiple layers in different ways, as long as they produce the same final function (after composing them, $\mathbf{f}_1 \circ \mathbf{f}_2 \circ \cdots \circ \mathbf{f}_L = \mathbf{f}$), and that they respect the causal dependencies provided by $\boldsymbol{I} + \boldsymbol{A}$. This is similar to the linear example, and what we used to design the generative model in Eq. 5 of §4.

**Collapsed SCM** Just as before, we can expand all the different layers and write a (probably complex) formula that encompasses all changes in a single step. It is easy to show that the composition of these functions has, in general, a Jacobian matrix structurally equivalent to $\boldsymbol{I} + \sum_{l}^{\text{diam}(\boldsymbol{A})} \boldsymbol{A}^l$. Specifically, their composition will be of the form $\prod_l (\boldsymbol{I} + \boldsymbol{A})$.

**Reverse SCM** Since we assume that each $\tilde{f}_i(\mathbf{x}_{\text{pa}_i}, \mathbf{u}_i)$ is bijective with respect to $\mathbf{u}_i$, we can always compute its inverse to obtain $\mathbf{u}_i$ as a function of the observed values, $\mathbf{u}_i = \tilde{f}_i^{-1}(\mathbf{x}_{\text{pa}_i}, \mathbf{x}_i)$. Clearly, its Jacobian matrix will be structurally equivalent to $\boldsymbol{I} + \boldsymbol{A} \equiv \boldsymbol{I} - \boldsymbol{A}$.

Therefore, we can always reason as we did with the linear case, but using Jacobian matrices to talk about causal dependencies between variables, and possibly having a complex and/or non-closed formulation of the generative/abductive functions.

# C  Do-operator: interventions and counterfactuals

## C.1  Definition and algorithms

In this section, we extend on the do-operator implementation described in §5, and provide the step-by-step algorithms to perform interventions (Alg. 1) and compute counterfactuals (Alg. 2).

**Semantics**  Recalling §5, the *do-operator* [30], denoted as $do(\mathrm{x}_i = \alpha)$, is defined as a mathematical operator that simulates a physical intervention on an SCM $\mathcal{M}$, inducing an alternative model $\mathcal{M}^{\mathcal{I}}$ that fixes the observational value $\mathrm{x}_i = \alpha$, and thus removes any causal dependency on $\mathrm{x}_i$. However, the definition does not describe the specifics on how to implement such an operation.

**Usual implementation**  Traditionally, we are given the recursive representation of an SCM (Fig. 8a), as discussed in App. B. As such, the do-operator $do(\mathrm{x}_i = \alpha)$ is usually carried out by replacing the $i$-th equation, i.e., the $i$-th component of $\tilde{\mathbf{f}}$, with a constant function. That is, by doing $\tilde{f}_i^{\mathcal{I}} := \alpha$. This yields an intervened SCM $\mathcal{M}^{\mathcal{I}} = (\tilde{\mathbf{f}}^{\mathcal{I}}, P_{\mathbf{u}})$ reflecting the data-generating process after such an intervention. Unfortunately, this implementation of the do-operator is quite specific to the recursive representation of the SCM (Fig. 4a), and does not translate well to the other equivalent representations discussed in §4 and App. B. The reason for this is that these representations compute the observational values x as a vector function of $\mathbf{u}$, without the iterative sampling process that goes through the intervened value that we replace. The only exception is the abductive model with knowledge on the causal graph and $L = 1$ (see §4), as it corresponds to the exact same case of Fig. 8d. If we have $L > 1$ however, whether this implementation works or not comes down to the specific implementation to compute the inverse of the network.

**Proposed implementation**  As discussed in §4, we instead propose to manipulate the SCM by modifying the exogenous distribution $P_{\mathbf{u}}$, while keeping the causal generator $\tilde{\mathbf{f}}$ untouched. Specifically, an intervention $do(\mathrm{x}_i = \alpha)$ updates $P_{\mathbf{u}}$, to have positive density mass on only those values that, when transformed to endogenous variables, the intervened variable yields the intervened value, $\mathrm{x}_i = \alpha$, while keeping the rest of distributions unaltered.

In other words, we define the intervened SCM as $\mathcal{M}^{\mathcal{I}} = (\tilde{\mathbf{f}}, P_{\mathbf{u}}^{\mathcal{I}})$, where the density of the updated distribution $P_{\mathbf{u}}^{\mathcal{I}}$ is of the form

$$p^{\mathcal{I}}(\mathbf{u}) \propto p(\mathbf{u}) \cdot \delta_{\{\tilde{\mathbf{f}}_i(\boldsymbol{x},\mathbf{u})=\alpha\}}(\mathbf{u}) \,, \tag{23}$$

and where the distributions of the rest of variables remain the same. Using the acyclic assumption, we know that the only way of altering the value of $\mathrm{x}_i$ without altering those of its parents is through $\mathrm{u}_i$ and, using the causal sufficiency assumption, we can squeeze the Dirac delta directly in the distribution of the $i$-th exogenous variable, such that:

$$p^{\mathcal{I}}(\mathbf{u}) = p_i^{\mathcal{I}}(\mathrm{u}_i|\mathbf{u}_{j\neq i}) \cdot \prod_{j\neq i} p_j(\mathrm{u}_j)\,, \quad \text{with} \quad p_i^{\mathcal{I}}(\mathrm{u}_i|\mathbf{u}_{j\neq i}) \propto p_i(\mathrm{u}_i) \cdot \delta_{\{\tilde{\mathbf{f}}_i(\boldsymbol{x},\mathbf{u})=\alpha\}}(\mathbf{u})\,. \tag{24}$$

In the case we consider, where all the generators are bijective given the parent nodes, the set $\delta_{\{\tilde{\mathbf{f}}_i(\boldsymbol{x},\mathbf{u})=\alpha\}}(\mathbf{u})$ contains a single element, and therefore in the main paper we simply write the $i$-th density as $p_i^{\mathcal{I}}(\mathrm{u}_i|\mathbf{u}_{j\neq i}) = \delta_{\{\tilde{\mathbf{f}}_i(\boldsymbol{x},\mathbf{u})=\alpha\}}(\mathbf{u})$. Note, as discussed in the main paper, that the density at this point should be positive, in other words, the element that yields $\alpha$ (and therefore $\alpha$) should be a plausible value.

Since this implementation does not make any assumption at all in the functional form of the generator, but directly works on the distribution of the exogenous variables, it can be implemented on any SCM representation (see Fig. 8 in App. B.1) and, hence, on any of the architectures for the causal NF in §4. Notice, however, that in order to properly work, that the data-generative process should be causally consistent (so that changes are properly propagated), and that it should properly isolate $\mathbf{u}$ (so that $\mathrm{u}_i$ accounts only for the stochasticity of $\mathrm{x}_i$ not explained by its parents).

**Algorithms**  The step-by-step algorithms to perform interventions and compute counterfactuals, using the described algorithm, are presented in Alg. 1 and Alg. 2, respectively. It follows the same description as the one given at the end of §5, and the only difference between both algorithms is the way that we obtain samples from the observed distribution (generated vs. given).

**Algorithm 1** Algorithm to sample from the interventional distribution, $P(\mathbf{x} \,|\, do(\mathrm{x}_i = \alpha))$.

1: **function** SAMPLEINTERVENEDDIST($i, \alpha$)
2:      $\mathbf{u} \sim P_{\mathbf{u}}$
3:      $\mathbf{x} \leftarrow T_{\boldsymbol{\theta}}^{-1}(\mathbf{u})$                       ▷ Sample a value from the observational distribution.
4:      $\mathrm{x}_i \leftarrow \alpha$                           ▷ Set $\mathrm{x}_i$ to the intervened value $\alpha$.
5:      $\mathrm{u}_i \leftarrow T_{\boldsymbol{\theta}}(\mathbf{x})_i$                         ▷ Change the $i$-th value of $\mathbf{u}$.
6:      $\mathbf{x} \leftarrow T_{\boldsymbol{\theta}}^{-1}(\mathbf{u})$
7:      **return** $\mathbf{x}$                           ▷ Return the intervened sample.
8: **end function**

---

**Algorithm 2** Algorithm to sample from the counterfactual distribution, $P(\mathbf{x}^{\mathrm{cf}} \,|\, do(\mathrm{x}_i = \alpha), \mathbf{x}^{\mathrm{f}})$.

1: **function** GETCOUNTERFACTUAL($\mathbf{x}^{\mathrm{f}}, i, \alpha$)
2:      $\mathbf{u} \leftarrow T_{\boldsymbol{\theta}}(\mathbf{x}^{\mathrm{f}})$                         ▷ Get $\mathbf{u}$ from the factual sample.
3:      $\mathrm{x}_i^{\mathrm{f}} \leftarrow \alpha$                          ▷ Set $\mathrm{x}_i$ to the intervened value $\alpha$.
4:      $\mathrm{u}_i \leftarrow T_{\boldsymbol{\theta}}(\mathbf{x}^{\mathrm{f}})_i$                     ▷ Change the $i$-th value of $\mathbf{u}$.
5:      $\mathbf{x}^{\mathrm{cf}} \leftarrow T_{\boldsymbol{\theta}}^{-1}(\mathbf{u})$
6:      **return** $\mathbf{x}^{\mathrm{cf}}$                        ▷ Return the counterfactual value.
7: **end function**

---

**Theoretical results**    Here, we briefly discuss way this method works, i.e., why the proposed implementation removes every dependency from the descendants with respect to the ancestors that go through the intervened value, as it is not directly obvious.

To see why, take the usual recursive representation of an SCM in the illustrative example from App. B.1 (Fig. 8a), and assume that we do $do(\mathrm{x}_2 = \alpha)$, where $\mathrm{x}_2 = 2\mathrm{x}_1 + \mathrm{u}_2$ in this example. By updating the density $p_2(\mathrm{u}_2)$, we have basically fixed the value of $\mathrm{u}_2$ to be the only one that keeps $\mathrm{x}_2 = \alpha$ given $\mathrm{x}_1$, i.e., $\mathrm{u}_2 = \alpha - 2\mathrm{x}_1$ (this can be clearly seen in Fig. 8d). If we now compute the dependency of $\mathrm{x}_3$ on $\mathrm{x}_1$, we get

$$\frac{\mathrm{d}\,\mathrm{x}_3}{\mathrm{d}\,\mathrm{x}_1} = \frac{\partial \mathrm{x}_3}{\partial \mathrm{x}_2}\frac{\mathrm{d}\,\mathrm{x}_2}{\mathrm{d}\,\mathrm{x}_1} = \frac{\partial \mathrm{x}_3}{\partial \mathrm{x}_2}\left(\frac{\partial \mathrm{x}_2}{\partial \mathrm{x}_1} + \frac{\partial \mathrm{x}_2}{\partial \mathrm{u}_2}\frac{\mathrm{d}\,\mathrm{u}_2}{\mathrm{d}\,\mathrm{x}_1}\right) = \frac{\partial \mathrm{x}_3}{\partial \mathrm{x}_2}\left(2 + 1\cdot(-2)\right) = 0\,. \tag{25}$$

In layman's terms, the value of $\mathrm{u}_2$ is chosen such that it fixes the value of $\alpha$, countering any influence that the parents could have on $\mathrm{x}_2$ (or any of its intermediate values), and consequently in any of its descendants.

The general case can be similarly proven. Suppose that we do $do(\mathrm{x}_2 = \alpha)$, and that we want to compute the indirect influence of an ancestor, $\mathrm{x}_1$, on a descendant, $\mathrm{x}_3$, passing through $\mathrm{x}_2$. Since we are fixing the value of $\mathrm{u}_2$ (the input of the network) to produce an observed value $\mathrm{x}_2$ (the output of the network) of $\alpha$, we can use implicit differentiation to compute the influence of $\mathrm{u}_1$ (and therefore $\mathrm{x}_1$) on $\mathrm{x}_2$ via $\mathrm{u}_2$:

$$\alpha = \mathrm{x}_2(\mathrm{u}_1, \mathrm{u}_2) \xRightarrow{\frac{\mathrm{d}\,\mathrm{u}_1}{}} 0 = \frac{\partial \mathrm{x}_2}{\partial \mathrm{u}_1} + \frac{\partial \mathrm{x}_2}{\partial \mathrm{u}_2}\frac{\mathrm{d}\,\mathrm{u}_2}{\mathrm{d}\,\mathrm{u}_1} \Rightarrow \frac{\partial \mathrm{x}_2}{\partial \mathrm{u}_2}\frac{\mathrm{d}\,\mathrm{u}_2}{\mathrm{d}\,\mathrm{u}_1} = -\frac{\partial \mathrm{x}_2}{\partial \mathrm{u}_1}\,, \tag{26}$$

and, similar to Eq. 25, any *indirect* influence of the ancestor, $\mathrm{u}_1$, on the descendant, $\mathrm{x}_3$, through this intermediate variable, $\mathrm{x}_2$, cancels out:

$$\frac{\partial \mathrm{x}_3}{\partial \mathrm{x}_2}\frac{\mathrm{d}\,\mathrm{x}_2}{\mathrm{d}\,\mathrm{u}_1} = \frac{\partial \mathrm{x}_3}{\partial \mathrm{x}_2}\left(\frac{\partial \mathrm{x}_2}{\partial \mathrm{u}_1} + \frac{\partial \mathrm{x}_2}{\partial \mathrm{u}_2}\frac{\mathrm{d}\,\mathrm{u}_2}{\mathrm{d}\,\mathrm{u}_1}\right) = \frac{\partial \mathrm{x}_3}{\partial \mathrm{x}_2}\cdot 0 = 0\,. \tag{27}$$

We have proven this result not only theoretically, but also empirically through the accurate interventions computed for all the experiments from §6 and App. D. Moreover, this lack of correlation between ancestors and descendants through the intervened variables is clearly shown in pair plots such as the ones in Figs. 12 and 13.

## C.2 Interventions in previous works

We now put our implementation of the do-operator (see §5 and App. C) into context, by describing how the methods compared in §6, namely CAREFL [16] and VACA [34], proposed to perform interventions with their models.

**CAREFL [16]** Two different algorithms were proposed to sample from an interventional distribution in CAREFL: i) a sequential algorithm which mimics the usual implementation of the do-operator with the recursive representation of the SCM; and ii) a parallel algorithm that samples the counterfactual in a single call. While the first algorithm works, the parallel one—which is the one actually implemented—only works when intervening on root nodes.

This second algorithm for $do(x_2 = \alpha)$ is described as follows (see Alg. 2 in [16]): i) sample $\mathbf{u}$ from $P_{\mathbf{u}}$; ii) set $u_i$ to the $i$-th value obtained by applying the flow, $T_{\boldsymbol{\theta}}$, to an observation with $x_i = \alpha$ and $x_j = 0$ for $j \neq i$; and iii) return the value obtained by $T_{\boldsymbol{\theta}}^{-1}$ with $\mathbf{u}$.

While this algorithm resembles the one we proposed, the proposed method does not have into account that the value of $u_i$ to fix $\alpha$ *does depend* on the observed values of its parents, which is clear by looking at the linear illustrative example from Fig. 8d. As a consequence, the algorithm only works when the node has no parents, which is why we replaced it by the one we proposed for the comparisons in §6 and App. D.

**VACA [34]** Based on GNNs, the approach for intervening on VACA is completely reminiscent to the traditional implementation. Specifically, the authors propose to sever those edges in every layer of the GNN whose endpoints fall in the path generating the intervened variable, so that the ancestors have no way to influence it by design.

While the previous statement is true: ancestors cannot influence the intervened variable nor its descendants, here we argue that this process would require us to "recalibrate" the model, as the middle computations after an intervention change in more complex ways than removing the ancestors from the equation, while keeping the rest unchanged.

To see this, consider the following non-linear triangle SCM:

$$
\begin{cases}
x_1 = u_1 \\
x_2 = x_1^2 u_2 \\
x_3 = 2x_1 + \frac{x_2}{x_1} + \frac{x_2}{x_1^2} + u_3
\end{cases}
\tag{28}
$$

which VACA could learn with two layers through the following operations:

$$
\begin{cases}
z_1 = u_1 \\
z_2 = u_1 u_2 \\
z_3 = u_1 + u_2 + u_3
\end{cases}
\qquad
\begin{cases}
x_1 = z_1 \\
x_2 = z_1 z_2 \\
x_3 = z_1 + z_2 + z_3
\end{cases}
\; .
\tag{29}
$$

Now, if we were to intervene with $do(x_2 = \alpha)$, the real SCM would yield:

$$
\begin{cases}
x_1 = u_1 \\
x_2 = \alpha \\
x_3 = 2x_1 + \frac{\alpha}{x_1} + \frac{\alpha}{x_1^2} + u_3
\end{cases}
\tag{30}
$$

while VACA would yield:

$$
\begin{cases}
z_1 = u_1 \\
z_2 = \alpha \\
z_3 = u_1 + \alpha + u_3
\end{cases}
\qquad
\begin{cases}
x_1 = z_1 \\
x_2 = \alpha \\
x_3 = z_1 + \alpha + z_3
\end{cases}
\Rightarrow
\begin{cases}
x_1 = u_1 \\
x_2 = \alpha \\
x_3 = 2x_1 + 2\alpha + u_3
\end{cases}
\; ,
\tag{31}
$$

where we can clearly see that the expression for $x_3$ is different. In contrast, our causal NF would keep the generator as it is, and set $u_2$ to $\alpha/x_1^2$, yielding the correct value.

# D   Experimental details and extra results

In this section, we complement the description of the experimental section from §6, and provide the reader with additional results in the following subsections. First, we describe the details common to every experiment, and delve into the specifics of each experiment in their respective subsections.

**Hardware**   Every individual experiment shown in this paper ran on a single CPU with $8\,\text{GB}$ of RAM. To run all experiments, we used a local computing cluster with an automatic job assignment system, so we cannot ensure the specific CPU used for each particular experiment. However, we know that every experiment used one of the following CPUs picked randomly given the demand when scheduled: AMD EPYC 7702 64-Core Processor, AMD EPYC 7662 64-Core Processor, Intel(R) Xeon(R) CPU E5-2698 v4 @ 2.20GHz, or Intel(R) Xeon(R) CPU E5-2690 v4 @ 2.60GHz.

**Training and evaluation methodology**   For every experiment, we generated with using synthetic SCM a dataset with $20\,000$ training samples, $2500$ validation samples, and $2500$ test samples. We ran every model for $1000$ epochs, and the results shown in the manuscript correspond to the test set evaluation at the last epoch. For the optimization, we used Adam [17] with an initial learning rate of $0.001$, and reduce the learning rate with a decay factor of $0.95$ when it reaches a plateau longer than $60$ epochs. For hyperparameter tuning, we always perform a grid search with similar budget, and select the best hyperparameter combination according to validation loss, reporting always results from the test dataset in the manuscript. Every experiment is repeated 5 times, and we show averages and standard deviations.

**Datasets**   This section provides all the information of the SCMs employed in the empirical evaluation of §6 of the main paper, and the following subsections. The exogenous variables always follow a standard normal distribution $\mathcal{N}(0, 1)$, except for LARGEBD, where a uniform distribution $\mathcal{U}(0, 1)$ is used instead. Subsequently, we define the 12 SCMs employed—encompassing both linear and non-linear equations—and we additionally provide their causal graph in Fig. 9.

Let us first define the softplus operation as $s(x) = \log\left(1.0 + e^x\right)$.

**3-CHAIN$_\text{LIN}$:**

$$\tilde{f}_1(\mathbf{u}_1) = \mathbf{u}_1 \tag{32}$$

$$\tilde{f}_2(\mathbf{x}_1, \mathbf{u}_2) = 10 \cdot \mathbf{x}_1 - \mathbf{u}_2 \tag{33}$$

$$\tilde{f}_3(\mathbf{x}_2, \mathbf{u}_3) = 0.25 \cdot \mathbf{x}_2 + 2 \cdot \mathbf{u}_3 \tag{34}$$

**3-CHAIN$_\text{NLIN}$:**

$$\tilde{f}_1(\mathbf{u}_1) = \mathbf{u}_1 \tag{35}$$

$$\tilde{f}_2(\mathbf{x}_1, \mathbf{u}_2) = e^{\mathbf{x}_1/2} + \mathbf{u}_2/4 \tag{36}$$

$$\tilde{f}_3(\mathbf{x}_2, \mathbf{u}_3) = \frac{(\mathbf{x}_2 - 5)^3}{15} + \mathbf{u}_3 \tag{37}$$

**4-CHAIN$_\text{LIN}$:**

$$\tilde{f}_1(\mathbf{u}_1) = \mathbf{u}_1 \tag{38}$$

$$\tilde{f}_2(\mathbf{x}_1, \mathbf{u}_2) = 5 \cdot \mathbf{x}_1 - \mathbf{u}_2 \tag{39}$$

$$\tilde{f}_3(\mathbf{x}_2, \mathbf{u}_3) = -0.5 \cdot \mathbf{x}_2 - 1.5 \cdot \mathbf{u}_3 \tag{40}$$

$$\tilde{f}_4(\mathbf{x}_3, \mathbf{u}_4) = \mathbf{x}_3 + \mathbf{u}_4 \tag{41}$$

**5-CHAIN$_\text{LIN}$:**

$$\tilde{f}_1(\mathbf{u}_1) = \mathbf{u}_1 \tag{42}$$

$$\tilde{f}_2(\mathbf{x}_1, \mathbf{u}_2) = 10 \cdot \mathbf{x}_1 - \mathbf{u}_2 \tag{43}$$

$$\tilde{f}_3(\mathbf{x}_2, \mathbf{u}_3) = 0.25 \cdot \mathbf{x}_2 + 2 \cdot \mathbf{u}_3 \tag{44}$$

$$\tilde{f}_4(\mathbf{x}_3, \mathbf{u}_4) = \mathbf{x}_3 + \mathbf{u}_4 \tag{45}$$

$$\tilde{f}_5(\mathsf{x}_4, \mathsf{u}_5) = -\mathsf{x}_4 + \mathsf{u}_5 \tag{46}$$

**COLLIDER**$_{\text{LIN}}$:

$$\tilde{f}_1(\mathsf{u}_1) = \mathsf{u}_1 \tag{47}$$

$$\tilde{f}_2(\mathsf{u}_2) = 2 - \mathsf{u}_2 \tag{48}$$

$$\tilde{f}_3(\mathsf{x}_1, \mathsf{x}_2, \mathsf{u}_3) = 0.25 \cdot \mathsf{x}_2 - 0.5 \cdot \mathsf{x}_1 + 0.5 \cdot \mathsf{u}_3 \tag{49}$$

**FORK**$_{\text{LIN}}$:

$$\tilde{f}_1(\mathsf{u}_1) = \mathsf{u}_1 \tag{50}$$

$$\tilde{f}_2(\mathsf{u}_2) = 2 - \mathsf{u}_2 \tag{51}$$

$$\tilde{f}_3(\mathsf{x}_1, \mathsf{x}_2, \mathsf{u}_3) = 0.25 \cdot \mathsf{x}_2 - 1.5 \cdot \mathsf{x}_1 + 0.5 \cdot \mathsf{u}_3 \tag{52}$$

$$\tilde{f}_4(\mathsf{x}_3, \mathsf{u}_4) = \mathsf{x}_3 + 0.25 \cdot \mathsf{u}_4 \tag{53}$$

**FORK**$_{\text{NLIN}}$:

$$\tilde{f}_1(\mathsf{u}_1) = \mathsf{u}_1 \tag{54}$$

$$\tilde{f}_2(\mathsf{u}_2) = \mathsf{u}_2 \tag{55}$$

$$\tilde{f}_3(\mathsf{x}_1, \mathsf{x}_2, \mathsf{u}_3) = \frac{4}{1 + e^{-\mathsf{x}_1 - \mathsf{x}_2}} - \mathsf{x}_2^2 + 0.5 \cdot \mathsf{u}_3 \tag{56}$$

$$\tilde{f}_4(\mathsf{x}_3, \mathsf{u}_4) = \frac{20}{1 + e^{0.5 \cdot \mathsf{x}_3^2 - \mathsf{x}_3}} + \mathsf{u}_4 \tag{57}$$

**LARGEBD**$_{\text{NLIN}}$: Let us define

$$\text{L}(x, y) = s(x + 1) + s(0.5 + y) - 3.0, \tag{58}$$

and let us call $\text{CDF}^{-1}(\mu, b, x)$ the quantile function of a Laplace distribution with location $\mu$, scale $b$, evaluated at $x$. Then the structural equations are

$$\tilde{f}_1(\mathsf{u}_1) = s(1.8 \cdot \mathsf{u}_1) - 1 \tag{59}$$

$$\tilde{f}_2(\mathsf{x}_1, \mathsf{u}_2) = 0.25 \cdot \mathsf{u}_2 + \text{L}(\mathsf{x}_1, 0) \cdot 1.5 \tag{60}$$

$$\tilde{f}_3(\mathsf{x}_1, \mathsf{u}_3) = \text{L}(\mathsf{x}_1, \mathsf{u}_3) \tag{61}$$

$$\tilde{f}_4(\mathsf{x}_2, \mathsf{u}_4) = \text{L}(\mathsf{x}_2, \mathsf{u}_4) \tag{62}$$

$$\tilde{f}_5(\mathsf{x}_3, \mathsf{u}_5) = \text{L}(\mathsf{x}_3, \mathsf{u}_5) \tag{63}$$

$$\tilde{f}_6(\mathsf{x}_4, \mathsf{u}_6) = \text{L}(\mathsf{x}_4, \mathsf{u}_6) \tag{64}$$

$$\tilde{f}_7(\mathsf{x}_5, \mathsf{u}_7) = \text{L}(\mathsf{x}_5, \mathsf{u}_7) \tag{65}$$

$$\tilde{f}_8(\mathsf{x}_6, \mathsf{u}_8) = 0.3 \cdot \mathsf{u}_8 + (s(\mathsf{x}_6 + 1) - 1) \tag{66}$$

$$\tilde{f}_9(\mathsf{x}_7, \mathsf{x}_8, \mathsf{u}_9) = \text{CDF}^{-1}\left(-s\left(\frac{\mathsf{x}_7 \cdot 1.3 + \mathsf{x}_8}{3} + 1\right) + 2, 0.6, \mathsf{u}_9\right) \tag{67}$$

**SIMPSON**$_{\text{NLIN}}$:

$$\tilde{f}_1(\mathsf{u}_1) = \mathsf{u}_1 \tag{68}$$

$$\tilde{f}_2(\mathsf{x}_1, \mathsf{u}_2) = s(1 - \mathsf{x}_1) + \sqrt{3/20} \cdot \mathsf{u}_2 \tag{69}$$

$$\tilde{f}_3(\mathsf{x}_1, \mathsf{x}_2, \mathsf{u}_3) = \tanh(2 \cdot \mathsf{x}_2) + 1.5 \cdot \mathsf{x}_1 - 1 + \tanh(\mathsf{u}_3) \tag{70}$$

$$\tilde{f}_4(\mathsf{x}_3, \mathsf{u}_4) = \frac{\mathsf{x}_3 - 4}{5} + 3 + \frac{1}{\sqrt{10}} \cdot \mathsf{u}_4 \tag{71}$$

**SIMPSON**$_{\text{SYMPROD}}$:

$$\tilde{f}_1(u_1) = u_1 \tag{72}$$

$$\tilde{f}_2(x_1, u_2) = 2 \cdot \tanh(2 \cdot x_1) + \frac{1}{\sqrt{10}} \cdot u_2 \tag{73}$$

$$\tilde{f}_3(x_1, x_2, u_3) = 0.5 \cdot x_1 \cdot x_2 + \frac{1}{\sqrt{2}} \cdot u_3 \tag{74}$$

$$\tilde{f}_4(x_1, u_4) = \tanh(1.5 \cdot x_1) + \sqrt{\frac{3}{10}} \cdot u_4 \tag{75}$$

**TRIANGLE**$_{\text{LIN}}$:

$$\tilde{f}_1(u_1) = u_1 + 1 \tag{76}$$

$$\tilde{f}_2(x_1, u_2) = 10 \cdot x_1 - u_2 \tag{77}$$

$$\tilde{f}_3(x_1, x_2, u_3) = 0.5 \cdot x_2 + x_1 + u_3 \tag{78}$$

**TRIANGLE**$_{\text{NLIN}}$:

$$\tilde{f}_1(u_1) = u_1 + 1 \tag{79}$$

$$\tilde{f}_2(x_1, u_2) = 2 \cdot x_1^2 + u_2 \tag{80}$$

$$\tilde{f}_3(x_1, x_2, u_3) = \frac{20}{1 + e^{-x_2^2 + x_1}} + u_3 \tag{81}$$

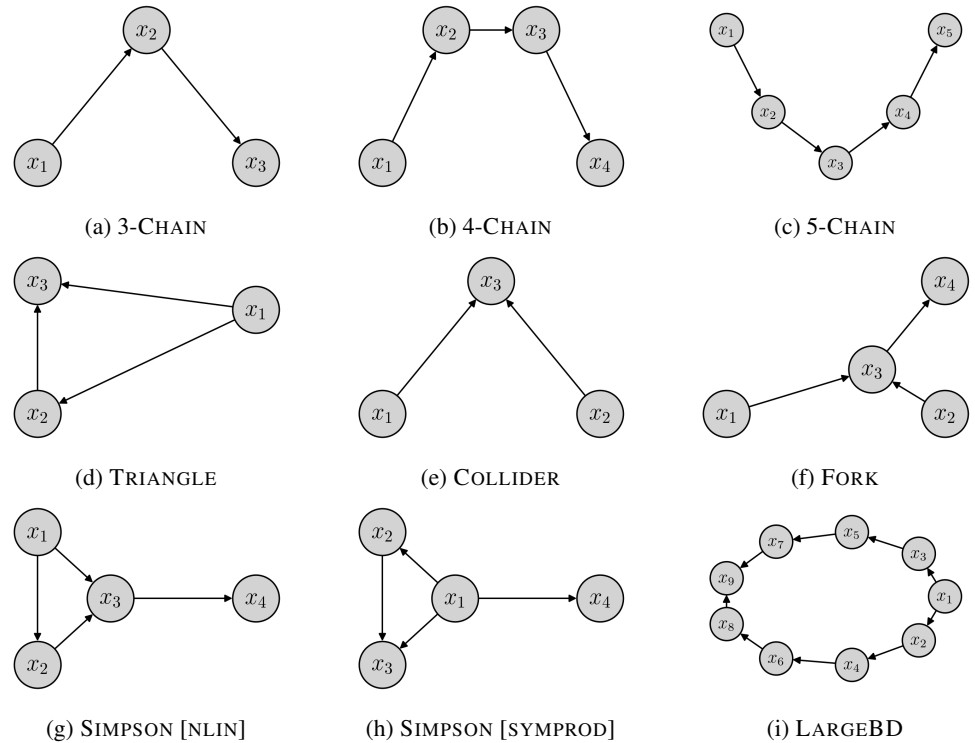

(a) 3-CHAIN     (b) 4-CHAIN     (c) 5-CHAIN

(d) TRIANGLE     (e) COLLIDER     (f) FORK

(g) SIMPSON [NLIN]     (h) SIMPSON [SYMPROD]     (i) LARGEBD

Figure 9: Causal graph of the different SCMs considered in §6 and App. D.

## D.1 Ablation: Time complexity

As the first additional ablation study, we evaluate the time complexity of the design choices introduced in §4. Fig. 10 summarizes the results, where the x-axis represents the number of nodes in the dataset,

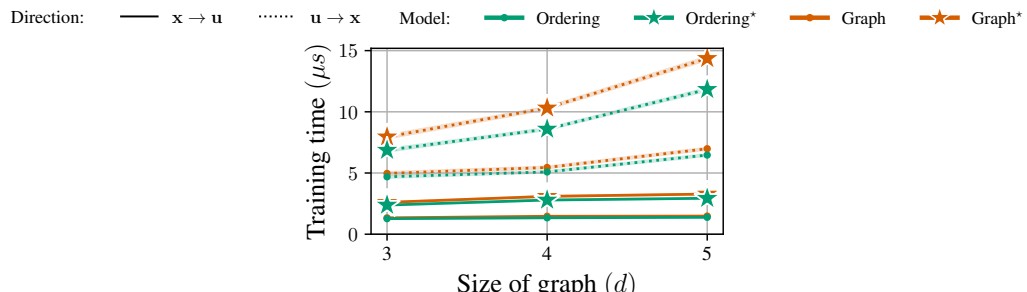

Figure 10: Time complexity comparison between the different design choices discussed in §4.

$d$, and the y-axis indicates the time (in microseconds) required for a single forward pass of the normalizing flow during training. We can find several interesting insights:

**Results**   First, abductive models ($\mathbf{x} \rightarrow \mathbf{u}$, solid lines) train significantly faster compared to generative models ($\mathbf{u} \rightarrow \mathbf{x}$, dotted lines). This emphasizes the computational cost associated with inverting autoregressive flows during training. Furthermore, it is worth noting that the time complexity of abductive models remains constant regardless of the number of nodes $d$, whereas the complexity of generative models increases linearly with $d$, as indicated in the right-most column of Tab 1. Note that we should see the exact opposite behaviour when sampling.

Second, the inclusion of Jacobian regularization, represented by a star marker, introduces a significant time overhead. This observation is clearly depicted in Fig. 10, where plot lines with the star marker are consistently above their counterparts.

Finally, leveraging causal graph information or relying solely on ordering (represented in orange and green, respectively) has minimal impact on computational time. This is once again clear in Fig. 10, as green and orange pairs largely coincide.

## D.2   Ablation: Base distribution

We now assess to which extent a mismatch of the distribution $P_{\mathbf{u}}$ between the SCM and the causal NF negatively affects performance. To this end, we consider two more complex SCMs—SIMPSON [11] and TRIANGLE [34]—and distributions—Normal and Laplace—for which we either fix or learn their parameters during training. Both SCMs use a standard Normal distribution for $P_{\mathbf{u}}$.

**Hyperparameter tuning**   While we fixed the flow to have a single MAF [24] layer with ELU [4] activation functions, we determined through cross-validation the optimal number of layers and hidden units of the MLP network within MAF. Specifically, we considered the following combinations ($[a, b]$ represents two layers with $a$ and $b$ hidden units): $[16, 16, 16, 16], [32, 32, 32], [16, 16, 16], [32, 32], [32], [64]$. As discussed at the start of the section, we report test results for the configuration with the best validation performance at the last epoch.

**Results**   The results, shown in Fig. 11a, reveal a notable distinction between Normal and Laplace distributions in terms of density estimation. However, this discrepancy appears to have minimal implications for Average Treatment Effect (ATE) and counterfactual estimation. We hypothesize that this disparity originates from dissimilarities in their tails, as it can be inferred by the slight edge of Normal over Laplace on the last column—which measures *per sample* differences—where bigger errors happen at the outliers elements which are, by definition, scarce. Interestingly, with this particular architecture of causal NF, every model struggles to model the denser TRIANGLE SCM.

## D.3   Ablation: Flow architecture

Considering the observed challenges faced by the Masked Autoregressive Flow (MAF) [24] layer in accurately modelling the TRIANGLE SCM in the previous experiment, we further investigate the potential impact of flow architecture on performance.

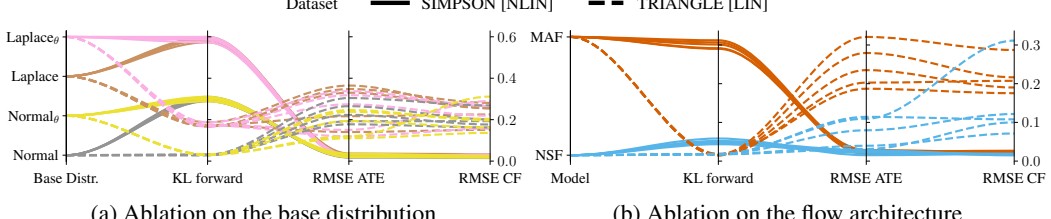

(a) Ablation on the base distribution       (b) Ablation on the flow architecture

Figure 11: Performance on the SIMPSON$_{\text{NLIN}}$ and TRIANGLE$_{\text{LIN}}$ datasets of causal NFs with a) different base distributions (Normal and Laplace), where $\theta$ indicates that we learn the parameters of the base distribution; and b) flow architectures. Differences in base distribution affect KL divergence, while the choice of flow architecture influences the overall performance.

**Hyperparameter Tuning**     We cross-validate again the optimal number of layers and hidden units of the MLP internally used by the unique layer of the Causal NF. We consider the following values ($[a, b]$ represents two layers with $a$ and $b$ hidden units): $[32, 32, 32]$, $[16, 16, 16]$, $[32, 32]$, $[32]$, and $[64]$. As before, test results are reported for the configuration that achieved the best performance on the validation set at the final epoch.

**Results**     Fig. 11b summarizes our results, where we consider a Causal NF with one MAF [24] layer (depicted in orange), and a Causal NF with a Neural Spline Flow (NSF) [9] as layer (depicted in blue). Note that, NSFs are built on top of MAFs and abandoned the realm of affine ANFs, and are thus expected to outperform MAFs in general. We employed the same set of SCMs as in App. D.2.

Our empirical analysis reveals that the NSF consistently outperforms the MAF across the three metrics: observational distribution (measured by the KL divergence), ATE estimation, and counterfactual estimation. Whilst expected, these findings highlight the practical implications of selecting an appropriate flow architecture, which should be taken into consideration by practitioners.

### D.4 Comparison: Extra non-linear SCMs

In this section, we complement the results from §6.2 and provide a more extensive comparison of the proposed Causal NF, along with CAREFL [16] and VACA [34], on additional datasets.

**Hyperparameter Tuning**     For VACA, we cross-validated the dropout rate with values $\{0.0, 0.1\}$, the GNN layer architecture with $\{\text{GIN}, \text{PNA}, \text{PNADisjoint}\}$, (see [34] for details), and the number of layers in the MLP prior to the GNN with choices $\{1, 2\}$. For CAREFL, we cross-validated the number of layers in the flow, $\{1, \text{diam}\, \boldsymbol{A}\}$, and the number of layers and hidden units in the MLP composing the flow layers (same format as before), $\{[16, 16, 16], [32, 32], [32], [64]\}$. For Causal NF, we used the abductive model with a single layer, and cross-validated the number of layers and hidden units in the MLP composing the layer of the flow with values $\{[16, 16, 16, 16], [32, 32, 32], [16, 16, 16], [32, 32], [32], [64]\}$. We report test results for the configuration with the best validation performance at the final epoch.

**Results**     Tab 4 shows the performance of each model for all the considered datasets, further validating the conclusions drawn in the main manuscript: the proposed Causal NF consistently outperforms both CAREFL and VACA in terms of performance and computational efficiency. The performance of VACA is notably inferior, and its computation time is significantly longer, primarily due to the complexity of graph neural networks (GNNs). Our Causal NF achieves similar performance to CAREFL in terms of observational fitting, while surpassing it on interventional and counterfactual estimation tasks. Additionally, Causal NF outperforms CAREFL in computational speed. This is to be expected since the optimal CAREFL architectures often have multiple layers, resulting in increased computation time. In contrast, Causal NF has a single layer, reducing computational complexity.

Fig. 12 qualitative proofs the effectiveness of the proposed Causal NF in accurately modelling both observational and interventional distributions for the SIMPSON$_{\text{NLIN}}$ dataset. In this plot, blue represent the real distribution/samples, while orange represents the ones generated by Causal NF. Fig. 12a clearly shows that the model successfully captured the correlations among all variables in the observational distribution. Furthermore, Fig. 12b displays the interventional distribution

Table 4: Comparison, on different SCMs, of the proposed Causal NF, VACA [34], and CAREFL [16] with the do-operator proposed in §5. Results averaged over five runs.

| Dataset | Model | Performance | | | Time Evaluation (µs) | | |
|---|---|---|---|---|---|---|---|
| | | KL | $ATE_{RMSE}$ | $CF_{RMSE}$ | Training | Evaluation | Sampling |
| 3-CHAIN LIN [34] | Causal NF | $\mathbf{0.00_{0.00}}$ | $\mathbf{0.05_{0.01}}$ | $\mathbf{0.04_{0.01}}$ | $\mathbf{0.41_{0.06}}$ | $\mathbf{0.48_{0.10}}$ | $\mathbf{0.76_{0.06}}$ |
| | CAREFL† | $\mathbf{0.00_{0.00}}$ | $0.20_{0.13}$ | $0.20_{0.09}$ | $0.68_{0.24}$ | $0.97_{0.33}$ | $1.94_{0.77}$ |
| | VACA | $4.44_{1.03}$ | $5.76_{0.07}$ | $4.98_{0.10}$ | $36.19_{1.54}$ | $28.33_{0.72}$ | $75.34_{4.58}$ |
| 3-CHAIN NLIN [34] | Causal NF | $\mathbf{0.00_{0.00}}$ | $\mathbf{0.03_{0.01}}$ | $\mathbf{0.02_{0.01}}$ | $\mathbf{0.52_{0.06}}$ | $\mathbf{0.56_{0.03}}$ | $\mathbf{1.02_{0.05}}$ |
| | CAREFL† | $\mathbf{0.00_{0.00}}$ | $0.05_{0.02}$ | $0.04_{0.02}$ | $\mathbf{0.60_{0.22}}$ | $0.84_{0.22}$ | $1.66_{0.41}$ |
| | VACA | $12.82_{1.00}$ | $1.54_{0.03}$ | $1.32_{0.02}$ | $39.45_{4.12}$ | $30.93_{2.30}$ | $84.36_{9.60}$ |
| 4-CHAIN LIN | Causal NF | $\mathbf{0.00_{0.00}}$ | $\mathbf{0.07_{0.02}}$ | $\mathbf{0.04_{0.01}}$ | $\mathbf{0.56_{0.08}}$ | $\mathbf{0.62_{0.15}}$ | $\mathbf{1.54_{0.40}}$ |
| | CAREFL† | $\mathbf{0.00_{0.00}}$ | $0.16_{0.07}$ | $0.14_{0.04}$ | $\mathbf{0.70_{0.28}}$ | $0.99_{0.20}$ | $2.85_{0.54}$ |
| | VACA | $13.14_{0.73}$ | $3.82_{0.01}$ | $3.72_{0.05}$ | $61.85_{5.06}$ | $49.31_{4.11}$ | $92.06_{7.93}$ |
| 5-CHAIN LIN | Causal NF | $0.01_{0.00}$ | $\mathbf{0.12_{0.02}}$ | $\mathbf{0.08_{0.01}}$ | $\mathbf{0.62_{0.19}}$ | $\mathbf{0.69_{0.15}}$ | $\mathbf{1.91_{0.44}}$ |
| | CAREFL† | $\mathbf{0.00_{0.00}}$ | $0.47_{0.23}$ | $0.46_{0.22}$ | $\mathbf{0.79_{0.41}}$ | $1.19_{0.25}$ | $4.21_{0.87}$ |
| | VACA | $17.31_{0.84}$ | $5.95_{0.05}$ | $6.06_{0.08}$ | $103.75_{10.04}$ | $80.81_{11.06}$ | $124.52_{20.86}$ |
| COLLIDER LIN [34] | Causal NF | $\mathbf{0.00_{0.00}}$ | $\mathbf{0.02_{0.01}}$ | $\mathbf{0.01_{0.00}}$ | $0.46_{0.12}$ | $0.56_{0.11}$ | $0.95_{0.19}$ |
| | CAREFL† | $\mathbf{0.00_{0.00}}$ | $\mathbf{0.02_{0.01}}$ | $\mathbf{0.01_{0.00}}$ | $\mathbf{0.39_{0.07}}$ | $\mathbf{0.45_{0.05}}$ | $\mathbf{0.74_{0.07}}$ |
| | VACA | $13.45_{0.43}$ | $0.22_{0.01}$ | $0.86_{0.02}$ | $37.22_{3.55}$ | $28.77_{4.22}$ | $71.21_{6.73}$ |
| FORK LIN [2] | Causal NF | $\mathbf{0.00_{0.00}}$ | $\mathbf{0.03_{0.01}}$ | $\mathbf{0.01_{0.00}}$ | $\mathbf{0.52_{0.05}}$ | $\mathbf{0.59_{0.08}}$ | $\mathbf{1.57_{0.57}}$ |
| | CAREFL† | $\mathbf{0.00_{0.00}}$ | $0.04_{0.01}$ | $0.02_{0.00}$ | $0.60_{0.17}$ | $0.78_{0.16}$ | $2.39_{1.06}$ |
| | VACA | $8.75_{0.73}$ | $0.87_{0.02}$ | $1.43_{0.02}$ | $45.84_{4.64}$ | $34.66_{2.39}$ | $73.29_{4.70}$ |
| FORK NLIN [2] | Causal NF | $\mathbf{0.00_{0.00}}$ | $\mathbf{0.07_{0.02}}$ | $\mathbf{0.07_{0.00}}$ | $0.63_{0.16}$ | $0.74_{0.31}$ | $1.84_{0.84}$ |
| | CAREFL† | $0.01_{0.01}$ | $0.11_{0.04}$ | $0.18_{0.07}$ | $\mathbf{0.57_{0.17}}$ | $\mathbf{0.77_{0.08}}$ | $\mathbf{1.96_{0.17}}$ |
| | VACA | $5.09_{0.60}$ | $2.01_{0.03}$ | $3.19_{0.06}$ | $49.22_{5.48}$ | $42.13_{2.95}$ | $101.02_{18.94}$ |
| LARGEBD NLIN [11] | Causal NF | $\mathbf{1.51_{0.04}}$ | $\mathbf{0.02_{0.00}}$ | $\mathbf{0.01_{0.00}}$ | $\mathbf{0.52_{0.10}}$ | $\mathbf{0.60_{0.17}}$ | $\mathbf{3.05_{0.66}}$ |
| | CAREFL† | $\mathbf{1.51_{0.05}}$ | $0.05_{0.01}$ | $0.08_{0.01}$ | $0.84_{0.47}$ | $1.18_{0.17}$ | $8.25_{1.29}$ |
| | VACA | $53.66_{2.07}$ | $0.39_{0.00}$ | $0.82_{0.02}$ | $164.92_{11.10}$ | $137.88_{15.72}$ | $167.94_{25.75}$ |
| SIMPSON NLIN [11] | Causal NF | $\mathbf{0.29_{0.01}}$ | $\mathbf{0.04_{0.01}}$ | $\mathbf{0.02_{0.00}}$ | $\mathbf{0.58_{0.18}}$ | $\mathbf{0.63_{0.26}}$ | $\mathbf{1.57_{0.64}}$ |
| | CAREFL† | $\mathbf{0.29_{0.01}}$ | $\mathbf{0.04_{0.01}}$ | $0.03_{0.00}$ | $0.69_{0.32}$ | $1.02_{0.26}$ | $2.95_{0.79}$ |
| | VACA | $18.97_{0.66}$ | $0.60_{0.00}$ | $1.19_{0.02}$ | $54.76_{7.46}$ | $43.69_{7.92}$ | $87.01_{16.33}$ |
| SIMPSON SYMPROD [11] | Causal NF | $\mathbf{0.00_{0.00}}$ | $\mathbf{0.07_{0.01}}$ | $\mathbf{0.12_{0.02}}$ | $0.59_{0.17}$ | $\mathbf{0.60_{0.11}}$ | $\mathbf{1.51_{0.30}}$ |
| | CAREFL† | $\mathbf{0.00_{0.00}}$ | $0.10_{0.02}$ | $0.17_{0.04}$ | $\mathbf{0.49_{0.15}}$ | $0.81_{0.19}$ | $1.91_{0.33}$ |
| | VACA | $13.85_{0.64}$ | $0.89_{0.00}$ | $1.50_{0.04}$ | $49.26_{4.09}$ | $37.78_{3.41}$ | $79.20_{14.60}$ |
| TRIANGLE LIN [34] | Causal NF | $\mathbf{0.00_{0.00}}$ | $0.24_{0.05}$ | $0.21_{0.05}$ | $\mathbf{0.54_{0.05}}$ | $\mathbf{0.56_{0.04}}$ | $\mathbf{1.05_{0.07}}$ |
| | CAREFL† | $\mathbf{0.00_{0.00}}$ | $\mathbf{0.15_{0.06}}$ | $\mathbf{0.14_{0.03}}$ | $0.60_{0.20}$ | $0.75_{0.05}$ | $1.50_{0.10}$ |
| | VACA | $3.82_{0.69}$ | $7.49_{0.07}$ | $7.22_{0.17}$ | $27.46_{1.53}$ | $21.61_{1.00}$ | $67.00_{6.23}$ |
| TRIANGLE NLIN [34] | Causal NF | $\mathbf{0.00_{0.00}}$ | $\mathbf{0.12_{0.03}}$ | $\mathbf{0.13_{0.02}}$ | $\mathbf{0.52_{0.07}}$ | $\mathbf{0.58_{0.07}}$ | $\mathbf{1.07_{0.12}}$ |
| | CAREFL† | $\mathbf{0.00_{0.00}}$ | $\mathbf{0.12_{0.03}}$ | $0.17_{0.03}$ | $0.57_{0.18}$ | $0.83_{0.26}$ | $1.68_{0.62}$ |
| | VACA | $7.71_{0.60}$ | $4.78_{0.01}$ | $4.19_{0.04}$ | $28.82_{1.21}$ | $23.00_{0.55}$ | $70.65_{3.70}$ |

obtained when we do $do(x_3 = -1.09)$, i.e., when we intervene on the 25-th empirical percentile of $x_3$. Remarkably, Causal NF accurately learns the distribution of descendant variables, i.e., $x_4$, and effectively breaks any dependency between the ancestors of the intervened variable and $x_4$. Additionally, Fig. 13 shows a similar analysis for 5-CHAIN$_{LIN}$, when we perform $do(x_3 = 2.18)$— which corresponds to intervening on the 75-th percentile of $x_3$—clearly showing that the correlations not involving the intervened path ($x_1 \rightarrow x_2$ and $x_4 \rightarrow x_5$) are preserved.

# E   Details on the fairness use-case

In this section, we provide additional details on the use-case of fairness auditing and classification using the German dataset [8], whose causal graph is shown in Fig. 14.

**Training**   For this section, we performed minimal hyperparameter tuning, and only tested a few combinations by hand. We decided to use a Neural Spline Flow (NSF) [9] for the single layer of the Causal NF, which internally uses an MLP with 3 layers, and 32 hidden units each. We use Adam [17] as the optimizer, with a learning rate of 0.01, along with a plateau scheduler with a decay factor of

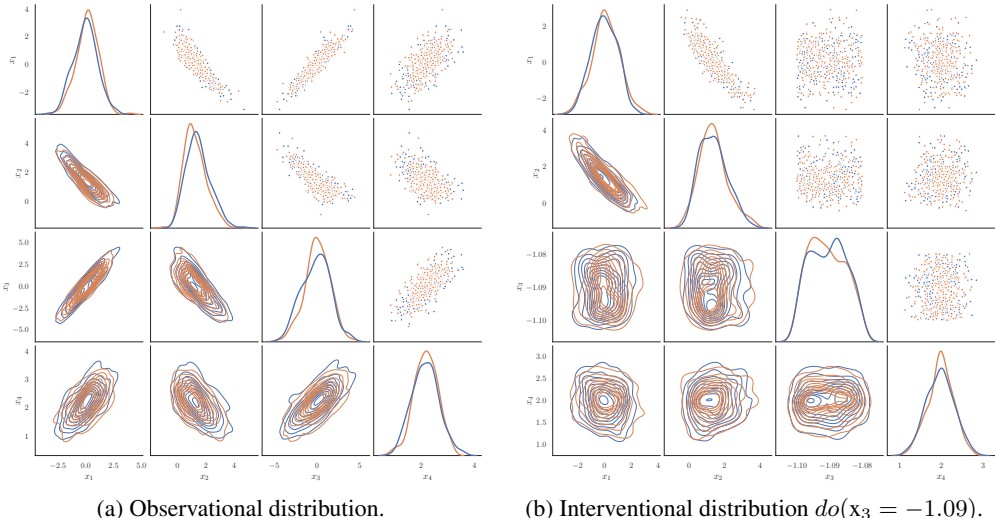

(a) Observational distribution.

(b) Interventional distribution $do(\mathrm{x}_3 = -1.09)$.

Figure 12: Pair plot of real (in blue) and generated (in orange) data of SIMPSON$_{\mathrm{NLIN}}$. On the left are samples from the true and learnt observational distribution. On the right are samples from the true and learnt interventional distribution when $do(\mathrm{x}_3 = -1.09)$. The plot illustrates that the dependency of $\mathrm{x}_4$ on the ancestors of $\mathrm{x}_3$, namely $\mathrm{x}_1$ and $\mathrm{x}_2$, is effectively broken.

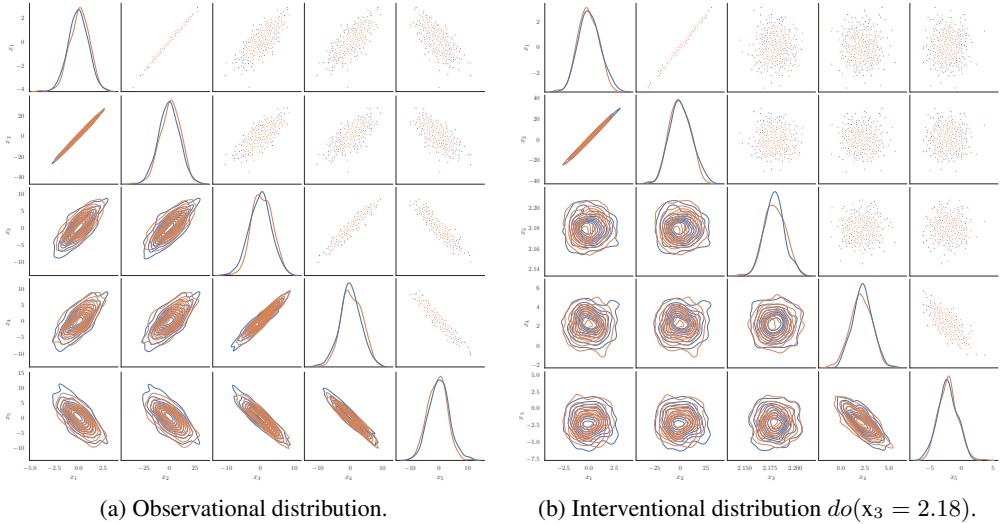

(a) Observational distribution.

(b) Interventional distribution $do(\mathrm{x}_3 = 2.18)$.

Figure 13: Pair plot of real (in blue) and generated (in orange) data of 5-CHAIN$_{\mathrm{LIN}}$. On the left are samples from the true and learnt observational distribution. On the right are samples from the true and learnt interventional distribution when $do(\mathrm{x}_3 = 2.18)$. The plot illustrates that the dependency of $\mathrm{x}_4$ and $\mathrm{x}_5$ on the ancestors of $\mathrm{x}_3$, namely $\mathrm{x}_1$ and $\mathrm{x}_2$, is effectively broken.

0.9 and a patience parameter of 60 epochs. The training is performed for 1000 epochs, and the results are reported using 5-fold cross-validation with a $80 - 10 - 10$ split for train, validation, and test data.

**Results** On addition to the results from §7, Fig. 15 shows two pair plots from one of the 5 runs, chosen at random. The true empirical distribution is shown in blue, and the learnt distribution by Causal NF is depicted in orange. Specifically, Fig. 15a illustrates the observational distribution, and Fig. 15b the interventional distribution, obtained when we intervene on the *sex* variable and set it to 1, i.e., $do(\mathrm{x}_1 = 1)$. We can observe that Causal NF achieves a remarkable fit in both cases, demonstrating its capability to handle discrete data, and partial knowledge of the causal graph.

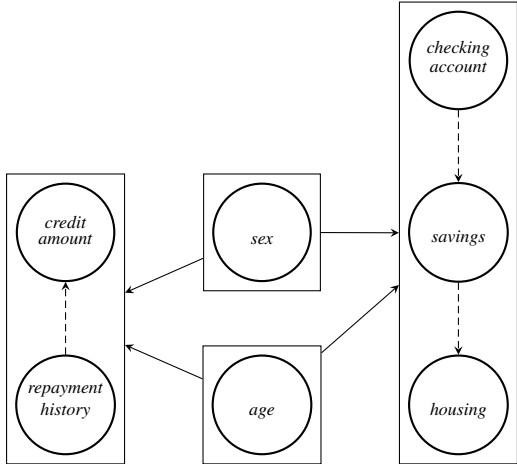

Figure 14: Partial causal graph used for the German Credit dataset [8]. Rectangles show the strongly connected components (SCCs) grouping different variables. Solid arrows represent causal relationships between SCCs, and dashed arrows represent an arbitrary order picked to learn the joint distribution of each SCC with an ANF. See App. A.2.2 for an in-depth explanation on the proposed method to deal with partial causal graphs using causal NFs.

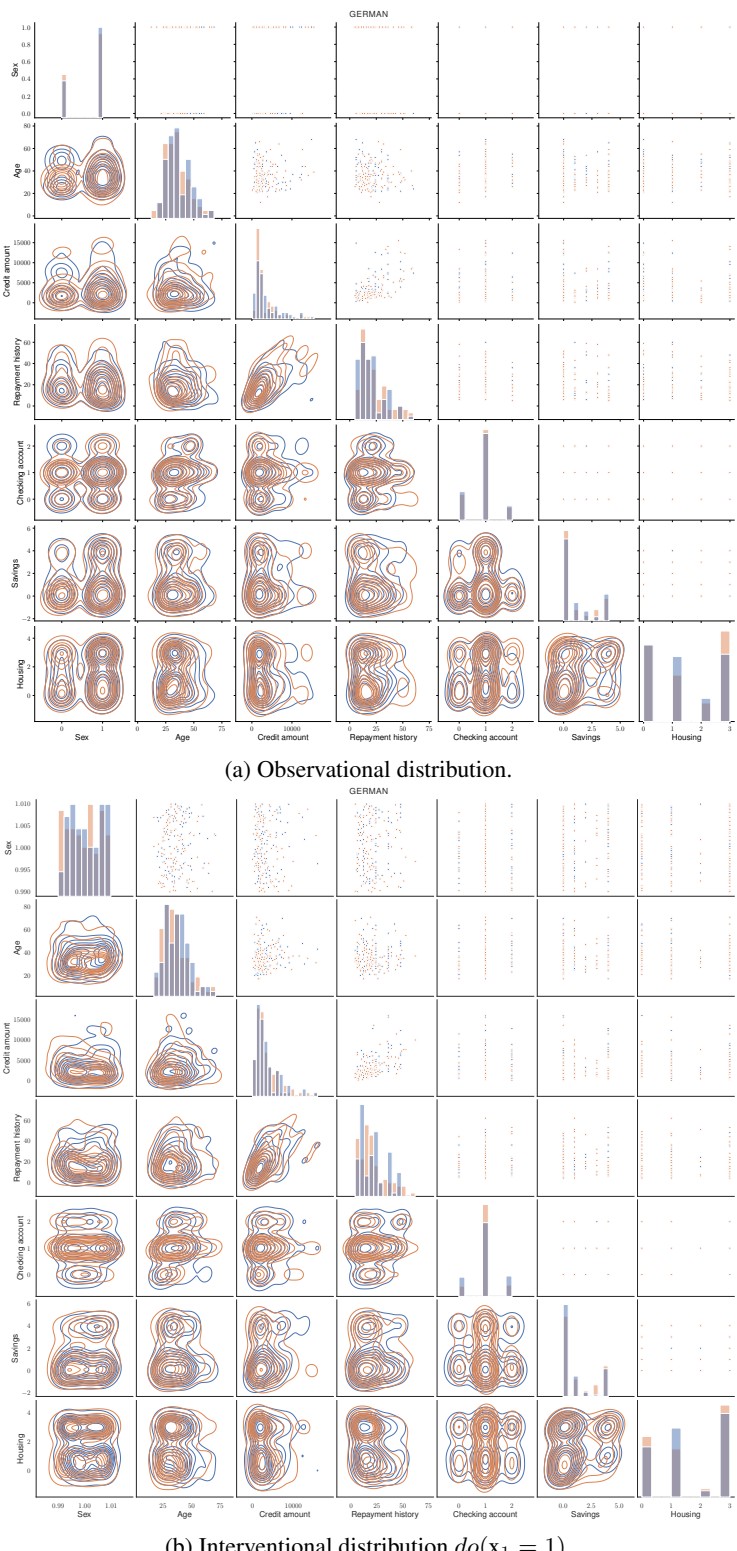

(a) Observational distribution.

(b) Interventional distribution $do(\mathrm{x}_1 = 1)$.

Figure 15: Pair plot of real (in blue) and generated (in orange) data of German dataset. Above, the samples from the true and learnt observational distributions. Below, the samples from the true and learnt interventional distributions when $do(\mathrm{x}_1 = 1)$. The plot illustrates that Causal NF is able to handle discrete data and correctly intervene.

