# OpenReview forum: "Causal normalizing flows: from theory to practice"
_NeurIPS.cc/2023/Conference — NeurIPS 2023 oral_

### Official Review · Reviewer_p91A · 2023-07-04

**Soundness:** 3 good
**Presentation:** 3 good
**Contribution:** 3 good
**Rating:** 7
**Confidence:** 3

**Summary:**

The authors propose a normalizing flow (NF) model that incorporates causal information, either via a partial order or complete DAG. They derive identifiability conditions for causal estimands and implement their algorithm on a range of simulated and real-world datasets. The resulting NF can be used to compute treatment effects and counterfactual probabilities, under some reasonable assumptions on the data generating process.

**Strengths:**

This is a strong contribution on an important topic. The manuscript is well-motivated and thoroughly researched. The writing is clear and the experimental validation is convincing. Despite a few minor questions/comments (see below), I am generally supportive of this work.

**Weaknesses:**

There are a few minor points that left me somewhat confused. I suspect these could be cleared up with a few brief lines.

-I’m unsure how to interpret $\sum_{n=1}^\infty \mathbf{G}^n$. Why the infinite sum? The examples in Fig. 4 all appear finite?


-It appears that the empirical evaluation only considers causal chains of the form $X_1 \rightarrow \dots \rightarrow X_d$. What about more complicated/realistic causal structures?


-If I understand Fig. 5 correctly, the ordering model (green curves) outperforms the graph model (orange curves) by KL-divergence and causal effect estimation in generative models when $L < 3$, while the ordering model dominates effect estimation across all numbers of layers in the abductive model. This echoes the results summarized in Table 1. I feel I must be missing something, however – why should a model do better with a causal ordering than it does with the entire graph? Surely the latter is strictly more informative?


**Questions:**

See above.

---

> ### Author Rebuttal · Authors · 2023-08-09
>
> We thank the reviewer for their valuable comments and for the supportive words. Please see our comments to the reviewer’s comments below:
>
> > I’m unsure how to interpret $\sum_{n=1}^\infty G^n$ . Why the infinite sum? The examples in Fig. 4 all appear finite?
>
> The short and honest answer is that the infinite is there to save space. Since $G$ is acyclic, that sum is always finite as $G^n$ becomes $0$ for $n > \operatorname{diam}(G)$, but we did not need to define any notation for the diameter of the graph anywhere else. However, we agree that it can be confusing to the reader, and we will rewrite it as a finite sum with $\operatorname{diam}(G)$ terms in the next revision of the manuscript.
>
> > It appears that the empirical evaluation only considers causal chains. What about more complicated/realistic causal structures?
>
> We only consider causal chains for the ablation study shown in the main paper, as using causal chains is easy to understand and sufficient to verify the claims we make in Section 4 with respect to the network design. Yet, we do consider more complex SCMs (in terms of causal graphs and structural equations) in Section 6 and Appendix D.4 (see Figure 9 for a reference on the considered graphs). Moreover,   the use-case in Section 7 shows a more realistic scenario with mixed-type data, and a partial causal graph (see Figure 14).
>
> > If I understand Fig. 5 correctly […] This echoes the results summarized in Table 1: Why should a model do better with a causal ordering than it does with the entire graph? Surely the latter is strictly more informative?
>
> Indeed, Figure 5 echoes Table 1. The goal of the ablation study in Section 6.1 is to empirically corroborate the theoretical findings summarized in Table 1 (e.g., that the generative model with the graph needs as many layers as the graph diameter), as well as to motivate the necessity of a network that is causally consistent by design. Since our theory only requires a causal ordering (Section 3), one could reasonably believe that any ANF would suffice, yet we empirically show that in practice we often fall on local optima with spurious correlations, and that a causally-consistent design effectively restricts the search space, easing the optimization process and, ultimately, yielding more accurate estimates of causal inference estimands.
>
> Given the positive assessment of our paper by the reviewer and, in case we have successfully clarified the questions above, we would appreciate it if  they could consider upgrading the score to better reflect the contributions of our work. Please refer to the [general comment](https://openreview.net/forum?id=QIFoCI7ca1&noteId=3m1mexttyb) for a summary of said contributions.

---

> > ### Comment · Reviewer_p91A · 2023-08-14
> > **Re: rebuttal**
> >
> > Many thanks to the authors for their comments and clarifications. I still believe my original score of 7 is fair and will keep it as such. Great work on this paper!

---

> > > ### Author Response · Authors · 2023-08-18
> > >
> > > We would like to thank the reviewer for reading our rebuttal and for the kind words, we _truly_ appreciate them.

---

### Official Review · Reviewer_azJz · 2023-07-06

**Soundness:** 3 good
**Presentation:** 3 good
**Contribution:** 4 excellent
**Rating:** 8
**Confidence:** 3

**Summary:**

n this paper, the authors derive and demonstrate the usefulness of normalizing flows (NF) in causal inference. As a first building block of their approach, the authors show that structural causal models (SCMs) can be expressed as triangular monotonic increasing maps.  By this reduction, they show that SCMs can be approximated by autoregressive NFs and that those are causally consistent.
They proceed by discussing different architectures to best model the SCM with NFs and conclude that an abductive setup for the NF might be the most suitable architecture.
In order to perform interventions on an NF they define the do-operator on those. Due to the recursive representation of their approach, the authors resort to defining an intervention by fixing the exogenous variables accordingly. In empirical evaluations, they show that their approach compares favourably to the SOTA on representing the true data distribution and causal inference tasks w.r.t. to both performance and time efficiency.
Lastly, they show on real-world data that their approach can be used for fairer, but still precise classification.

**Strengths:**

- The elaboration of the connections between SCMs, TMIs, and ANFs is well executed and can find additional applications in other research.
- The implementation of interventions as performing them on the exogenous variables is non-trivial and opens up further paths of investigation.
- Figures 2-4 are very valuable for presentation and understanding.
- Clear and sound experimental setup.
- Only mild and standard assumptions.
- Showcasing the usefulness of causal NFs in ML w.r.t. fairness.

**Weaknesses:**

- The paper would benefit of an example of how a learned causal NF behaves, how the interventions influence the output, and how it matches an SCM.
- Consider pushing some of the information from the appendix to the main text. E.g.:
	- 152: A sketch of the proof would strengthen the line of argumentation of this work.
	- 327: At least a short sentence on what this data is about.
- Some background on TMI maps could be introduced to make it easier to follow the content.
- No resources such as codebase, trained models, and analysis implementation are provided.

**Questions:**

No questions

**Limitations:**

The authors address the limitations of the work clearly and describe how their approach relies on their assumptions. Furthermore, they describe how violating each of their assumptions would affect the overall approach. These observations seem to be complete w.r.t. the possible limitations of this approach.

---

> ### Author Rebuttal · Authors · 2023-08-09
>
> We thank the reviewer for their valuable feedback, detailed summary, and positive comments. Please see our comments regarding the raised weaknesses below:
>
> > The paper would benefit of an example of how a learned causal NF behaves, how the interventions influence the output, and how it matches an SCM.
>
> We appreciate the comment, and we would appreciate more specific details on said example to properly address it during the rebuttal. To our understanding, the reviewer may be referring to something similar to the pair-plots we provide in Figures 12, 13, and 15 from the appendix, but we are happy to add other examples.
>
> > Consider pushing some of the information from the appendix to the main text. E.g.: …
>
> We thank the reviewer for the specific feedback, and we will implement these changes in the updated manuscript.
>
> > Some background on TMI maps could be introduced.
>
> We agree with the reviewer that more background on TMI maps can be helpful to the reader, and we will add it to the next revision of our work.
>
> > No resources such as codebase, trained models, and analysis implementation are provided.
>
> We would like to point out that we provide the code to reproduce the experiments of our work with the submission, in the same zip file where the appendix was found. We did not consider necessary providing the model weights, given how quick they are to train and run in all our experiments. In addition, we will make all the code necessary to reproduce all our experiments publicly available with the publication of our paper.
>
> As for the “analysis implementation”, we are unsure what the reviewer refers to exactly. Could the reviewer clarify, please? We could happily address any additional comments they may have.
>
> In case we have successfully clarified the points raised by the reviewer, we would appreciate it if  they could consider upgrading the score of our paper accordingly. Please refer to the [general comment](https://openreview.net/forum?id=QIFoCI7ca1&noteId=3m1mexttyb) for a summary of the main contributions of our work.

---

> > ### Comment · Reviewer_azJz · 2023-08-22
> >
> > Thanks authors for the informative reply. Indeed, the Figures 12, 13, and 15 from the appendix show what I would have liked to see in the main text (or a version of it), such that the effect of the interventions is visualized.
> >
> > As for codebase ect. there was a technical issue on my side which omitted those from me. So contrary to my previous assessment, these resources are indeed provided and good.
> >
> > With "analysis implementation" I intended the scripts in which you perform your analysis. These are provided in your supplementary material as well.
> >
> > Considering the information about the reproducibility of this work, I would like to raise my score to strong accept.

---

### Official Review · Reviewer_yYim · 2023-07-07

**Soundness:** 2 fair
**Presentation:** 3 good
**Contribution:** 2 fair
**Rating:** 6
**Confidence:** 2

**Summary:**

This paper proposes causal normalizing flows, a method to learn a structural causal model (SCM) using normalizing flows (NFs), when the causal order (and potentially more information about the graph) is available. They consider a general SCM where we have observed variables X which follow the equations X_i = f_i(X_pa(i), u_i) where pa(i) are causal parents of i and u_i is endogenous noise. Then, by unrolling the recurisve definition, the authors propose to model this distribution via NFs. In fact, because of acyclicity, the NFs are actually Triangular Monotonic Increasing (TMI) maps. Under the following assumptions
- The functions f_i are diffeomorphic
- the causal graph is acyclic
- the causal ordering is known (this can be a strong assumption, see below)
- causal sufficiency, i.e. the endogenous variables u_i are mutually independent
it follows froms prior works that the model is identifiable (means there is a unique product distribution and TMI maps given the observational dataset). The authors restate this result and use it as a basis for their experimental explorations.

The main selling point of the work is that with approproate design choices of causal normalizing flows, the causal task of do-operations can be performed efficiently. Different design choices for causal normalizing flows are proposed, including generative (modeling the mixing directly) and abductive (modeling the inverse map) methods. To handle do-operations, the authors propose modifying the endogenous variable directly (since the recursive SCM form is lost). They also show how to adapt their framework when the data is discrete or partial knowledge of the graph is available. For experiments, ablation studies compare the design choices, and the framework is also compared to baselines CAREFL and VACA, showing that CNFs perform better on a few SCM tasks. Finally, a fairness use-case is shown with the German credit dataset. The target audience are people interested in causal inference.

**Strengths:**

- Both structural causal modeling and normalizing flows (by now) are well-studied topics and this work is a nice interplay between these two frameworks.

- The application to deduce credit risk while being unbiased on sex is an interesting and a bit unusual application of their framework.

**Weaknesses:**

- Assuming knowledge of causal ordering is usually a very strong requirements in applications, since if we have the knowledge of causal ordering, we can use traditional nonlinear regression techniques, e.g. with spline functions, to learn the underlying SCMs. Therefore, this may potentially be a very limiting assumption in experiments and reduce the usefulness of the proposed method.

- The identifiability results are not novel and therefore the thrust of this work is experimental. As the authors clarify, theorem 1 is taken from Xi and Bloem-Reddy.

**Questions:**

Please comment on the issues above.

**Limitations:**

Limitations have been discussed.

---

> ### Author Rebuttal · Authors · 2023-08-09
>
> We thank the reviewer for their valuable comments, as well as the highly detailed summary of our work. Before commenting on the concerns raised by the reviewer, we would like to clarify a few points that may have been overlooked or misunderstood based on the review’s summary:
>
> > [Identifiable] means there is a unique product distribution and TMI maps given the observational dataset.
>
> This definition would rather correspond to strong identifiability, which only occurs if the base distributions of the flow and SCM are the same. Otherwise, we obtain weak identifability, where the mapping is not unique, but differs on a component-wise transformation (Theorem 1), as depicted in Figure 3, which is sufficient to enable causal inference.
>
> > … showing that CNFs perform better on a few SCM tasks.
>
> Based on the “few” above, we are unsure whether this passed unnoticed, but we would like to remind that in Appendix D.4 we provide results for 12 different SCMs. We restricted this number to 3 in the main manuscript due to space constraints.
>
> Next, regarding the reviewer’s concerns:
>
> > Assuming knowledge of causal ordering is usually a very strong requirements in applications, since if we have the knowledge of causal ordering, we can use traditional nonlinear regression techniques, e.g. with spline functions, to learn the underlying SCMs.
>
> We would like to highlight that we focus in our paper in causal inference, i.e., recovering the value of the exogenous variables to then obtain  estimates for interventional and counterfactual queries. For such tasks, and in contrast to causal discovery problems, it is indeed common to assume access to the causal graph in addition to observational data [[1](https://aaai.org/papers/08159-vaca-designing-variational-graph-autoencoders-for-causal-queries/), [2](http://arxiv.org/abs/2302.00860), [3](https://openreview.net/forum?id=vouQcZS8KfW), [4](https://proceedings.mlr.press/v177/sanchez22a.html), [5](http://arxiv.org/abs/2109.04173)], when no other assumptions are considered (e.g., access to interventional data [[6]](https://proceedings.mlr.press/v202/nasr-esfahany23a.html) or to auxiliary variables [[7]](https://proceedings.mlr.press/v108/khemakhem20a.html)).  Importantly, to the best of our understanding, traditional nonlinear regression techniques for causal inference iteratively  fit a  regressor (e.g., spline function) per node, conditioned on its parents. As a consequence, in addition to assume a known causal graph, parameters are not amortized and errors tend to propagate along causal paths (from root nodes to leave nodes).
>
> Yet, as we acknowledge that access to the full causal graph may be restrictive in real-world applications, we provide an algorithm to group variables together if only partial knowledge on the causal ordering is available (the description in lines 171-178 are expanded in Appendix A.2.2).
>
>
>
>
> > The identifability results are not novel and therefore the thrust of this work is experimental.
>
> We respectfully and strongly disagree with this comment by the reviewer. While we do adapt the theory from Xi and Bloem-Reddy [[8]](https://proceedings.mlr.press/v206/xi23a.html) to the problem of causal inference, we would like to stress that their work involves identifiability of latent variable models in an ICA framework, where there is **not any causal consideration.**  Our main contribution lies in the fact that by casting SCMs and ANFs as TMI maps, we can re-formulate a causal inference problem (i.e., recovering the true exogenous variables)  as a blind-source separation problem, and thus leverage their identifiability results. To the best of our knowledge, our work is the first to provide identifiability guarantees for such a broad family of SCMs (lines 110-113). For ease of exposition, we restated Proposition 5.2 from [[8]](https://proceedings.mlr.press/v206/xi23a.html) to match our setting, which may lead to confusions. We will make this more clear in the next revision of the manuscript.
>
> Besides the  aforementioned identifiability result, the characterization of causally-consistent ANF network designs, an efficient implementation of the do-operator suitable for causal NFs, as well as extensions to mixed-type data and partial causal graphs are also theoretical and novel contributions of our work. We make these points more precise in the [general comment](https://openreview.net/forum?id=QIFoCI7ca1&noteId=3m1mexttyb).
>
>
> We hope to have addressed the main concerns, as well as clarified potential misunderstandings, raised by the reviewer, and thus would  appreciate it if the reviewer could consider upgrading the score of our paper accordingly. Please refer to the [general comment](https://openreview.net/forum?id=QIFoCI7ca1&noteId=3m1mexttyb) for a summary of the main contributions of our work, as well as for a summary of the changes planned for our paper.
>
> ---
>
> [1] [VACA: Design of Variational Graph Autoencoders for Interventional and Counterfactual Queries](https://aaai.org/papers/08159-vaca-designing-variational-graph-autoencoders-for-causal-queries/)
>
> [2] [Interventional and Counterfactual Inference with Diffusion Models](http://arxiv.org/abs/2302.00860)
>
> [3] [Neural Causal Models for Counterfactual Identification and Estimation](https://openreview.net/forum?id=vouQcZS8KfW)
>
> [4] [Diffusion Causal Models for Counterfactual Estimation](https://proceedings.mlr.press/v177/sanchez22a.html)
>
> [5] [Relating Graph Neural Networks to Structural Causal Models](http://arxiv.org/abs/2109.04173)
>
> [6] [Counterfactual Identifiability of Bijective Causal Models](https://proceedings.mlr.press/v202/nasr-esfahany23a.html)
>
> [7] [Variational Autoencoders and Nonlinear ICA: A Unifying Framework](https://proceedings.mlr.press/v108/khemakhem20a.html)
>
> [8] [Indeterminacy in Generative Models: Characterization and Strong Identifiability](https://proceedings.mlr.press/v206/xi23a.html)

---

> > ### Comment · Reviewer_yYim · 2023-08-15
> > **Response to rebuttal**
> >
> > I thank the authors for their response. With their clarifications, I upgraded my score.

---

> > > ### Author Response · Authors · 2023-08-18
> > >
> > > We thank the reviewer for reading our rebuttal and updating their score accordingly. If there is anything else we could help with during the rebuttal, we will be happy to do so.

---

### Official Review · Reviewer_515Z · 2023-07-10

**Soundness:** 3 good
**Presentation:** 3 good
**Contribution:** 3 good
**Rating:** 7
**Confidence:** 3

**Summary:**

This work explores the use of normalizing flows for causal reasoning. The authors demonstrate that causal models can be identified from observational data using autoregressive normalizing flows. They discuss design choices, learning strategies, and the implementation of the do-operator to handle interventional and counterfactual questions. Through experiments, they validate their approach, compare it to alternative methods, and show its effectiveness in addressing real-world problems with mixed discrete-continuous data and partial causal graph knowledge.

**Strengths:**

Novelty: This work introduces a novel approach by employing causal normalizing flows to identify the underlying causal ordering and effectively address interventional and counterfactual queries using the do-operator. To the best of my knowledge, this methodology is both novel and reasonable.

Significance: The ability to predict the interventional effect of the causal data-generating process is a highly important problem with practical implications.

Contribution: The paper presents a clear and comprehensive method, accompanied by necessary conditions that support the theoretical aspects. The overall technical contribution is commendable.

The writing quality of this paper is good, and it provides ample experimental results to validate its effectiveness. I thoroughly enjoyed reading this paper.

**Weaknesses:**

Challenge: From my understanding, the previous work has already achieved satisfactory results regarding the identifiability of causal normalizing flows. This paper extends these findings to include counterfactual and interventional reasoning. While the theoretical contribution and methodology are commendable, I think this extension is relatively straightforward, and the challenge presented may not be substantial. Consequently, this could be considered the primary weakness of the paper.

**Questions:**

I don't have any specific questions about the main paper, but I recommend that the authors consider transferring some crucial conclusions from the supplementary material to the main paper.

**Limitations:**

.

---

> ### Author Rebuttal · Authors · 2023-08-09
>
> We thank the reviewer for their positive comments, and we are flattered to know that the reviewer had a good time reading our work. While there are no specific questions, we would like to add a few clarifications below that we hope helps the reviewer better contextualize our work:
>
> > From my understanding, the previous work has already achieved satisfactory results regarding the identifability of causal normalizing flows.
>
> We are unsure to which previous work the reviewer refers exactly:
>
> 1. If the reviewer refers to the paper by Xi and Bloem-Reddy [[1]](https://proceedings.mlr.press/v206/xi23a.html), from which we adapt Theorem 1, note that their work involves identifiability of latent variable models in an ICA framework, where there is **no causal consideration.** Thus, our main contribution lies in the fact that by casting SCMs and ANFs as TMI maps, we can re-formulate causal inference (i.e., recovering the true exogenous variables)  as a blind-source separation problem, and thus leverage their identifiability results. To the best of our knowledge, our work is the first to provide identifiability guarantees for such a broad family of SCMs (lines 110-113).
> 2. If the reviewer refers instead to the work that introduced CAREFL [[2]](http://proceedings.mlr.press/v130/khemakhem21a.html), there are a number of important differences worth pointing out:
>     1. The main focus of CAREFL is causal discovery (i.e., estimating the causal graph), while ours is performing causal inference (given a causal ordering/graph).  These are two related but different problems, each with their own challenges and applications.  Importantly, the main theoretical result of CAREFL regards only the bivariate case (each one being uni-dimensional) with Gaussian exogenous variables.
>     2. Their work assumes affine ANFs, while we allow for any type of ANF. For example, NSFs  [[3]](https://proceedings.neurips.cc/paper/2019/hash/7ac71d433f282034e088473244df8c02-Abstract.html) (which we use in the fairness use-case, Section 7) are non-affine ANFs.
>     3. In a similar note, they assume additive-noise SCMs, which is a subset of the bijective SCMs assumed in our work.
> That said, we acknowledge that CAREFL indeed hints a connection between ANFs and SCMs, which in this work we: i) properly formalize, generalize, and make this connection precise; ii) provide general identifiability results; and iii) propose a network design to efficiently learn the underlying SCM.
>
> > I think this extension is relatively straightforward, and the challenge presented may not be substantial.
>
> We respectfully disagree with the reviewer here. While we agree that the final design is straightforward (this is in our opinion a positive outcome of our work), the contributions and challenges should not be overlooked. Specifically, in addition to the novelty of our identifiability results (see our answer on relation to Xi and Bloem-Reddy [[1]](https://proceedings.mlr.press/v206/xi23a.html) above), we first show in Corollary 2  that causal consistency is a necessary condition at the global optima; and then, in Section 4, introduce the necessary conditions to design a  causally consistent ANF, which needs to: i) use the graph information, ii) have a single layer, and iii) be defined from $x$ to $u$. Any other of the possible considered networks will not be causally consistent by design, and thus may lead to poor estimates of interventional and counterfactual queries. Moreover, as acknowledged by reviewer azJz,  the proposed implementation of the do-operator is also non-trivial.
>
>
> > I recommend that the authors consider transferring some crucial conclusions from the supplementary material to the main paper.
>
> We thank the reviewer for the suggestion, and we will make space to move important remarks back to the main manuscript. Reviewer azJz made a similar comment with specific content to add, and there may be some overlap. We have summarized the changes for the next revision in the [general comment](https://openreview.net/forum?id=QIFoCI7ca1&noteId=3m1mexttyb). If the reviewer has in mind specific parts of the Appendix to be moved to the main paper, please let us know so that we can also address them.
>
> We would appreciate it if the reviewer could confirm if we have successfully addressed their main comments and thus could consider upgrading the score of our paper accordingly. Please refer to the [general comment](https://openreview.net/forum?id=QIFoCI7ca1&noteId=3m1mexttyb) for a summary of the main contributions of our work.
>
> ---
>
> [1] [Indeterminacy in Generative Models: Characterization and Strong Identifiability](https://proceedings.mlr.press/v206/xi23a.html)
>
> [2] [Causal Autoregressive Flows](http://proceedings.mlr.press/v130/khemakhem21a.html)
>
> [3] [Neural Spline Flows](https://proceedings.neurips.cc/paper/2019/hash/7ac71d433f282034e088473244df8c02-Abstract.html)

---

> > ### Comment · Reviewer_515Z · 2023-08-14
> > **response to authors**
> >
> > Thank you for the detailed responses.

---

> > > ### Comment · Reviewer_515Z · 2023-08-18
> > > **Clear accept**
> > >
> > > Having carefully read all the discussions, I decided to elevate my rating to a clear acceptance (7).

---

> > > > ### Author Response · Authors · 2023-08-18
> > > >
> > > > We are very thankful to the reviewer for their invested effort to revisit all the discussion about our paper and for the positive feedback.

---

### Official Review · Reviewer_1x9i · 2023-07-17

**Soundness:** 4 excellent
**Presentation:** 3 good
**Contribution:** 4 excellent
**Rating:** 8
**Confidence:** 4

**Summary:**

This work explores the use of normalizing flows (NFs) in causal inference. The authors demonstrate that causal models can be identified from observational data using autoregressive NFs. They investigate design choices and implement the do-operator in causal NFs to answer interventional and counterfactual questions. The experiments validate their approach and show that causal NFs can effectively address real-world problems with mixed data types and partial knowledge of the causal graph.

**Strengths:**

The authors clearly stated the problem of interest as well as compare with relevant previously showed approaches. They also conducts extensive experiments, including ablation analysis. Finally, they discussed practical limitations and present possible lines of investigation as future work.

**Weaknesses:**

The number and characteristics of the given dataset are limited when compared to practical applications. The ablation analysis miss hyperparameter tuning.



**Questions:**

The authors should give the intuition of presenting Fig. 1 in a more clearly way.

It is well-known that computational complexity is a challenge when using normalizing flows, especially for high-dimensional data. In this matter, I believe that would be difficult to use the proposed method with such data (e.g. images). Is it limited to tabular data? Is it possible to use it with high-dimensional data? Any pre-processing steps are necessary?

It would be viable to learn the causal graph (e.g. PC) and use it as input the proposed method? My concern is whether it would be wast of time, given that the information is available in the data. In other words, is there a necessity of domain knowledge to use the proposed method?

**Limitations:**

The datasets used are not strictly representative of practical applications. I believe that there is an avenue of possible developments here turn the proposed method largely applicable.

---

> ### Author Rebuttal · Authors · 2023-08-09
>
>
> We thank the reviewer for their valuable feedback, which points out details that we will clarify in the updated manuscript. Please see our responses addressing the specific concerns below:
>
> > The number and characteristics of the given dataset are limited when compared to practical applications.
>
> We would like to point out that we test our framework in 12 synthetic SCMs (Table 4, App. D.4), including diverse causal graphs with both linear and non-linear structural equations (see Figure 9 in App. D). In addition,  we also use a real-world dataset, the German Credit,  which contains mixed-type data and partial knowledge on the causal graph. Our empirical evaluation is on par, in terms of number and complexity of the considered SCMs, with the one in VACA [[1](https://aaai.org/papers/08159-vaca-designing-variational-graph-autoencoders-for-causal-queries/)], and we believe it is sufficient to show the validity of our theory motivating design choices for causal normalizing flows. We agree with the reviewer that there is an avenue of future works  applying the proposed  causal normalizing flows to many application domains, also involving large dimensional data.
>
> > The ablation analysis misses hyperparameter tuning.
>
> We are unsure if we fully understand the reviewer comment, but we would like to clarify that the main goal of our ablation study is to validate the claims with respect to the network design in Section 4. Regarding hyperparameter tuning, we test the choice of base distribution for the flow in Appendix D.2, as well as the impact of choosing a different flow architecture in Appendix D.3. Additionally, we provide details on hyperparameter tuning for all experiments in App. D.
>
>  > The authors should give the intuition of presenting Fig. 1 in a more clearly way.
>
> Thanks for the feedback. We indeed were over-concise in our explanation, and will update it with the following: "This is exemplified in Fig. 1, where our proposed framework is able to estimate the (unobserved) causal effect of externally intervening on the sensitive attribute $s$, using solely observed data (blue distribution) and partial information about the causal relationship between features."
>
> > Is it limited to tabular data? Is it possible to use it with high-dimensional data? Any pre-processing steps are necessary?
>
> In principle there is no limitation on the proposed causal normalizing flow that prevents us from applying it to high-dimensional data. That said, we would like to clarify that our proposed method is tied to the current limitations of normalizing flows and to the extent to which we can define a (potentially partial) causal graph. For example, NFs have been applied in image processing, but to the best of our knowledge are not the state of the art, and at the same time it does not seem sensible to build a causal graph at the level of pixels. However, one could consider using an image as a node in our causal NF, for example in medical applications where the image (e.g., X-ray) is one more dimension in the patient (potentially causal) data.
>
> > It would be viable to learn the causal graph? Is there a necessity of domain knowledge to use the proposed method?
>
> While the focus of our paper is not causal discovery, we believe that our work could also be extended for causal discovery. For example, one idea would be to integrate the proposed causal NF into the DECI  framework [[2]](http://arxiv.org/abs/2202.02195), i.e., perform causal discovery by modelling the joint distribution of both the observed data and the causal graph as $P(G, X) = P(G) P(X|G)$, where the causal NF is used to fit $P(X|G)$. Note, however, that DECI only applies to structurally identifiable SCMs, specifically to continuous non-linear additive noise models (ANMs). In general, it is not possible to estimate a causal graph solely from observational data, and either domain knowledge or additional assumptions are needed.
>
> ---
>
> [1] [VACA: Design of Variational Graph Autoencoders for Interventional and Counterfactual Queries](https://aaai.org/papers/08159-vaca-designing-variational-graph-autoencoders-for-causal-queries/)
>
> [2] [Deep End-to-end Causal Inference](http://arxiv.org/abs/2202.02195)

---

### Author Rebuttal · Authors · 2023-08-09

We thank all the reviewers for their useful comments, which will ultimately help us improve the manuscript. We also want to thank them for their reassuring words towards our work. To name just a few, reviewers acknowledged our effort towards good writing and readability:
> I thoroughly enjoyed reading this paper - Reviewer 515Z

>  Figures 2-4 are very valuable for presentation and understanding - Reviewer azJz

the interdisciplinary character of our work:

> This work is a nice interplay between these two frameworks - Reviewer yYim

as well as its novelty and importance:

> This is a strong contribution on an important topic - Reviewer p91A

Additionally, to help during the rebuttal, we would like this general response to serve as a summary of the contributions of our work, as well as a summary of changes to the next updated manuscript.

**List of contributions**

The contributions of our work can be summarized as follows:

1. We formalize the connection between SCMs and ANFs by rewriting them as members of the same family of data-generating processes (i.e., TMI maps with factorized distributions).

2. Under this family, we rephrase causal inference problems as a specific instance of ICA, for which we can adapt the identifiability results from Xi and Bloem-Reddy [[2]](https://proceedings.mlr.press/v206/xi23a.html).

3. We demonstrate in Corollary 2 that, under the conditions of Theorem 1, being causally consistent is a necessary condition at the global optima when learning causal NFs. We bring such theoretical result into practice by discussing in Section 4 (and empirically validating it in Section 6.1) how to design and learn causally-consistent NFs.

4. We provide a new implementation of the do-operator that is well-suited for any SCM representation beyond its usual recursive formulation, enabling causal inference for the proposed causal NFs. We demonstrate theoretically that this is a proper implementation of the do-operator in App. C, and empirically in App. D.4.

5. We extend the above results to the cases of mixed-type data (App. A.2.1) and partial knowledge on the causal graph/ordering (App. A.2.2) to account for more realistic scenarios and thus make causal NFs applicable in a wide range of domains.

6. Finally, we empirically validate all our findings in 12 datasets (see Section 6 and Appendix D), and demonstrate its potential outreach with a fairness use-case in Section 7.

**Relation to previous work**

First, we would like to highlight that we indeed bring the identifiability results on ICA from Xi and Bloem-Reddy [[2]](https://proceedings.mlr.press/v206/xi23a.html) into the causal setting. We consider this contribution novel and significant as the original paper does not have any relationship to causality but only considers latent variable models.

Second, while CAREFL [[1]](http://proceedings.mlr.press/v130/khemakhem21a.html) focuses on causal discovery, it has hinted a connection between ANFs and SCMs. However, we are the first to: i) properly formalize and generalize (to a broader class of SCMs) this connection using TMI mappings; ii) provide identifiability results for causal NFs; and iii) propose a causal NF network design that is causally-consistent and, hence, can accurately and efficiently learn the underlying SCM generating the data.


**List of changes**

Following the reviewers' suggestions, we will carry out the following changes:

1. We will more clearly explain the intuition behind Figure 1. See response to reviewer 1x9i.
2. We will clarify that Theorem 1 has been re-stated from Xi and Bloem-Reddy [[2]](https://proceedings.mlr.press/v206/xi23a.html) to match our particular setting and ease exposition.
3. We will provide further intuition on the advantage of an abductive network vs. a generative one. In short, causal dependencies from $x$ to $u$ (parents) are sparser than those from $u$ to $x$ (ancestors). See response to reviewer 515Z.
4. We will revise the appendix and push important conclusions back to the manuscript. Of course, we are open to suggestions during the rebuttal. As suggested by reviewer azJz, we will push the following information to the main paper:
    1. A sketch of the proof for Corollary 2 to strengthen the line of argumentation of our work.
    2. A few lines introducing the German Credit dataset in more detail to the reader.
    3. Background on TMI maps. If this text becomes too lengthy, we will write a separate section in the appendix.
5. To improve readability, we will substitute the infinite sums $\sum_{n=1}^\infty G^n$ by the equivalent $\sum_{n=1}^{\operatorname{diam}(G)} G^n$. See response to reviewer p91A.
6. We will apply external feedback and fix typos and small errors. E.g.:
    1. In line 113 we call $u$ the endogenous variables, instead of exogenous.
    2. In the linear examples, we use the letter $G$ for the causal adjacency matrix and the actual linear operation, which can lead to confusions.

---

[1] [Causal Autoregressive Flows](http://proceedings.mlr.press/v130/khemakhem21a.html)

[2] [Indeterminacy in Generative Models: Characterization and Strong Identifiability](https://proceedings.mlr.press/v206/xi23a.html)

---

### Decision · Program_Chairs · 2023-09-21

**Decision:**

Accept (oral)

**Comment:**

This work explores the use of normalizing flows in causal inference. Assuming that data has been generated from a causal model and we know the causal graph as well as the causal ordering, the paper presents causal NFs and shows how to implement the do-operator in causal normalzing flows. This is a timely topic, and all reviewers suggest to accept the paper. I fulls support this.